# Human-Aligned Calibration for AI-Assisted Decision Making

**Nina L. Corvelo Benz**
Max Planck Institute for Software Systems
ETH Zürich
ninacobe@mpi-sws.org

**Manuel Gomez Rodriguez**
Max Planck Institute for Software Systems
manuel@mpi-sws.org

## Abstract

Whenever a binary classifier is used to provide decision support, it typically provides both a label prediction and a confidence value. Then, the decision maker is supposed to use the confidence value to calibrate how much to trust the prediction. In this context, it has been often argued that the confidence value should correspond to a well calibrated estimate of the probability that the predicted label matches the ground truth label. However, multiple lines of empirical evidence suggest that decision makers have difficulties at developing a good sense on when to trust a prediction using these confidence values. In this paper, our goal is first to understand why and then investigate how to construct more useful confidence values. We first argue that, for a broad class of utility functions, there exist data distributions for which a rational decision maker is, in general, unlikely to discover the optimal decision policy using the above confidence values—an optimal decision maker would need to sometimes place more (less) trust on predictions with lower (higher) confidence values. However, we then show that, if the confidence values satisfy a natural alignment property with respect to the decision maker's confidence on her own predictions, there always exists an optimal decision policy under which the level of trust the decision maker would need to place on predictions is monotone on the confidence values, facilitating its discoverability. Further, we show that multicalibration with respect to the decision maker's confidence on her own predictions is a sufficient condition for alignment. Experiments on four different AI-assisted decision making tasks where a classifier provides decision support to real human experts validate our theoretical results and suggest that alignment may lead to better decisions.

## 1 Introduction

In recent years, there has been an increasing excitement on the potential of machine learning models to improve decision making in a variety of high-stakes domains such as medicine, education or criminal justice [1–3]. One of the main focus has been binary classification tasks, where a classifier helps a decision maker by predicting a binary label of interest using a set of observable features [4–7]. For example, in medical treatment, the classifier may help a doctor by predicting whether a patient may benefit from a treatment. In college admissions, it may help an admissions committee by predicting whether a candidate may successfully complete an undergraduate program. In loan decisions, it may help a bank by predicting whether a prospective customer may default on a loan. In all these scenarios, the decision maker—the doctor, the committee or the bank—aim to use these predictions, together with their own predictions, to take good decisions that maximize a given utility function. In this context, since the predictions are unlikely to always match the truth, it has been widely agreed that the classifier should also provide a confidence value together with each prediction [8, 9].

37th Conference on Neural Information Processing Systems (NeurIPS 2023).

While the conventional wisdom is that the confidence value should be a well calibrated estimate of the probability that the predicted label matches the true label [10–16], multiple lines of empirical evidence have recently shown that decision makers have difficulties at developing a good sense on when to trust a prediction using these confidence values [17–19]. Therein, Vodrahali et al. [17] have shown that, in certain scenarios, decision makers take better decisions using uncalibrated probability estimates rather than calibrated ones. However, a theoretical framework explaining this puzzling observation has been missing and it is yet unclear what properties we should be looking for to guarantee that confidence values are useful for AI-assisted decision making. In our work, we aim to bridge this gap.

**Our contributions.** We start by formally characterizing AI-assisted decision making using a structural causal model (SCM) [20], as seen in Figure 1. Building upon this characterization, we first argue that, if a decision maker is rational, the level of trust she places on predictions will be monotone on the confidence values—she will place more (less) trust on predictions with higher (lower) confidence values. Then, we show that, for a broad class of utility functions, there are data distributions for which a rational decision maker can never take optimal decisions using calibrated estimates of the probability that the predicted label matches the true label as confidence values. However, we further show that, if the confidence values a decision maker uses satisfy a natural alignment property with respect to the confidence she has on her own predictions, which we refer to as human-alignment, then the decision maker can both be rational and take optimal decisions. In addition, we demonstrate that human-alignment can be achieved via multicalibration [11], a statistical notion introduced in the context of algorithmic fairness. In particular, we show that multicalibration with respect to the decision maker's confidence on her own predictions is a sufficient condition for human-alignment. Finally, we validate our theoretical framework using real data from four different AI-assisted decision making tasks where a classifier provides decision support to human decision makers in four different binary classification tasks. Our results suggest that, comparing across tasks, classifiers providing human-aligned confidence values facilitate better decisions than classifiers providing confidence values that are not human-aligned. Moreover, our results also suggest that rational decision makers' trust level increases monotonically with the classifier's provided confidence.

**Further related work.** Our work builds upon a rapidly increasing literature on AI-assisted decision making (refer to Lai et al. [21] for a recent review). More specifically, it is motivated by several empirical studies showing that decision makers have difficulties at modulating trust using confidence values [17–19], as discussed previously. In this context, it is also worth noting that other empirical studies have analyzed how other factors such as model explanations and accuracy modulate trust [22–26]. However, except for a very recent notable exception [27], theoretical frameworks, which could be used to better understand the mixed findings found by these empirical studies, have been missing. More broadly, our work also relates to a flurry of recent work on reinforcement learning with human feedback [28–30], which aims to better align the outputs of large language models (LLMs) with human preferences. However, our formulation is fundamentally different and our technical contributions are orthogonal to theirs.

## 2 A Causal Model of AI-Assisted Decision Making

We consider an AI-assisted decision making process where, for each realization of the process, a decision maker first observes a set of features $(x, v) \in \mathcal{X} \times \mathcal{V}$, then takes a binary decision $t \in \{0, 1\}$ informed by a classifier's prediction $\hat{y} = \operatorname{argmax}_y f_y(x)$, as well as confidence $f_{\hat{y}}(x) \in [0, 1]$, of a binary label of interest $y \in \{0, 1\}$, and finally receives a utility $u(t, y) \in \mathbb{R}$. Such an AI-assisted decision making process fits a variety of real-world applications. For example, in medical treatment, the features $(x, v)$ may comprise multiple sources of information regarding a patient's health[1], the label $y$ may indicate whether a patient would benefit from a specific treatment, the decision $t$ may indicate whether the doctor applies the specific treatment to the patient, and the utility $u(t, y)$ may quantify the trade-off between health benefit to the patient and economic cost to the decision maker.

In what follows, rather than working with both $\hat{y}$ and $f_{\hat{y}}(x)$, we will work with just $b = f_1(x)$, which we will refer to as classifier's confidence, without loss of generality[2]. Moreover, we will assume that

---

[1]Our formulation allows for a subset of the features $v$ to be available only to the decision maker but not to the classifier.

[2]We can recover $\hat{y}$ and $f_{\hat{y}}(x)$ from $b$, i.e., if $b > 0.5$, we have that $\hat{y} = 1$ and $f_{\hat{y}}(x) = b$; if $b < 0.5$, $\hat{y} = 0$ and $f_{\hat{y}}(x) = 1 - b$.

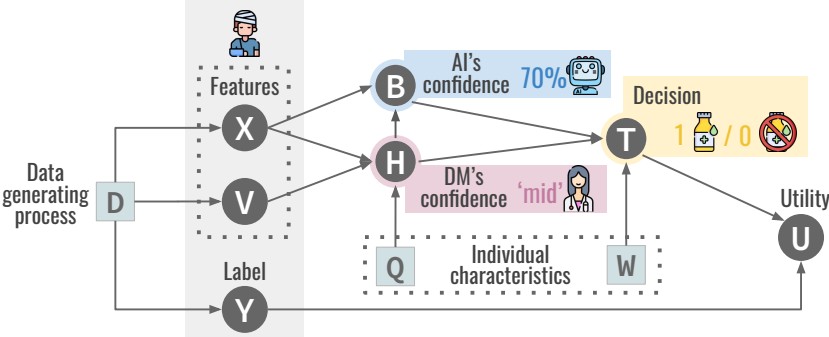

Figure 1: Our structural causal model $\mathcal{M}$. Orange circles represent endogenous random variables and blue boxes represent exogenous random variables. The value of each endogenous variable is given by a function of the values of its ancestors in the structural causal model, as defined by Eqs. 2 and 3. The value of each exogenous variable is sampled independently from a given distribution.

the utility $u(t, y)$ is greater if the value of $t$ and $y$ coincide, *i.e.*,

$$u(1,1) > u(1,0), \quad u(1,1) > u(0,1), \quad u(0,0) > u(1,0), \quad \text{and} \quad u(0,0) \geq u(0,1), \quad (1)$$

a condition that we think it is natural under an appropriate choice of label and decision values. For example, in medical diagnosis, if $t = 1$ means the patient is tested early for a disease and $y = 1$ means the patient suffers the disease, the above condition implies that the utility of either testing a patient who suffers the disease or not testing a patient who does not suffer the disease are greater than the utility of either not testing a patient who suffers the disease or testing a patient who does not suffer the disease. In condition 1, we allow for a non-strict inequality $u(0,0) \geq u(0,1)$ because, in settings in which the label $Y$ is only realized whenever the decision $t = 1$ (*e.g.*, in our previous example on medical treatment, we can only observe if a treatment is eventually beneficial or not if the patient is treated), it has been argued that, whenever $t = 0$, any choice of utility must be independent of the label value [4–6], *i.e.*, $u(0,0) = u(0,1) = u(0)$.

Next, we characterize the above AI-assisted decision making process using a structural causal model (SCM) [20], which we denote as $\mathcal{M}$. The SCM $\mathcal{M}$ is defined by a set of assignments, which entail a distribution $P^{\mathcal{M}}$ and divide naturally into two subsets. One subset comprises the features and the label[3], *i.e.*,

$$X = f_X(D) \quad V = f_V(D) \quad \text{and} \quad Y = f_Y(D), \quad (2)$$

where $D$ is an independent exogenous random variable, often called exogenous noise, characterizing the data generating process and $f_X$, $f_V$ and $f_Y$ are given functions[4]. The second subset comprises the decision maker and the classifier, *i.e.*,

$$H = f_H(X, V, Q), \quad B = f_B(X, H), \quad T = \pi(H, B, W) \quad \text{and} \quad U = u(T, Y), \quad (3)$$

where $f_H$ and $f_B$ are given functions, which determine the decision maker's confidence $H$ and classifier's confidence $B$ that the value of the label of interest is $Y = 1$, $\pi$ is a given AI-assisted decision policy, which determines the decision maker's decision $T$, $u$ is a given utility function, which determines the utility $U$, and $Q$ and $W$ are independent exogenous variables modeling the decision maker's individual characteristics influencing her own confidence $H$ and her decision $T$, respectively. By distinguishing both sources of noise, we allow for the presence of uncertainty on the decision $T$ even after conditioning on fixed confidence values $h$ and $b$. This accounts for the fact that, in reality, a decision maker may take different decisions $T$ for instances with the same confidence values $h$ and $b$. For example, in medical treatment, for two different patients with the same confidence $h$ and $b$, a doctor's decision may differ due to limited resources.

In our SCM $\mathcal{M}$, the decision maker's confidence $H$ refers to the confidence the decision maker has that the label $Y = 1$ *before* observing the classifier's confidence $B$. Moreover, following previous behavioral studies showing that human's confidence $H$ is discretized in a few distinct levels [32, 33], we assume $H$ takes values $h$ from a totally ordered discrete set $\mathcal{H}$. We say that the decision maker's

---

[3]We denote random variables with capital letters and realizations of random variables with lower case letters.
[4]Our model allows both for causal and anticausal features [31].

confidence $f_H$ is monotone (with respect to the probability distribution $P(Y = 1)$) if, for all $h, h' \in \mathcal{H}$ such that $h \leq h'$, it holds that $P(Y = 1 \mid H = h) \leq P(Y = 1 \mid H = h')$. Further, we allow the classifier's confidence $B$ to depend on the decision maker's confidence $H$ because this will be necessary to achieve human-alignment via multicalibration in Section 5. However, our negative result in Section 3 also holds if the classifier's confidence $f_B(X, H) = f_B(X)$ only depends on the features, as usual in most classifiers designed for AI-assisted decision making. In the remainder, we will use $Z = (X, H)$ and denote the space of features and human confidence values as $\mathcal{Z} = \mathcal{X} \times \mathcal{H}$. Figure 1 shows a visual representation of our SCM $\mathcal{M}$.

Under this characterization, we argue that, if a rational decision maker has decided $t$ under confidence values $b$ and $h$, then she would have decided $t' \geq t$ had the confidence values been $b' \geq b$ and $h' \geq h$ while holding "everything else fixed" [20]. For example, in medical treatment, assume a doctor's and a classifier's confidence that a patient would benefit from treatment is $b = h = 0.7$ and the doctor decides to treat the patient, then we argue that, if the doctor is rational, she would have treated the patient had the doctor's and the classifier's confidence been $b' = h' = 0.8 > 0.7$. Further, we say that any AI-assisted decision policy $\pi$ that satisfies this property is monotone, *i.e.*,

**Definition 1** (Monotone AI-assisted decision policy)**.** *An AI-assisted decision policy $\pi$ is monotone if and only if, for any $b, b' \in [0, 1]$ and $h, h' \in \mathcal{H}$ such that $b \leq b'$ and $h \leq h'$, it holds that $\pi(h, b, w) \leq \pi(h', b', w)$ for any $w \sim P^{\mathcal{M}}(w)$.*

Finally, note that, under any monotone AI-assisted decision policy, it trivially follows that

$$\mathbb{E}[T \mid H = h, B = b] \leq \mathbb{E}[T \mid H = h', B = b'], \tag{4}$$

where the expectation is over the uncertainty on the decision maker's individual characteristics and the data generating process.

## 3 Impossibility of AI-Assisted Decision Making Under Calibration

In AI-assisted decision making, classifiers are usually demanded to provide calibrated confidence values [10–16]. A confidence function $f_B : \mathcal{Z} \rightarrow [0, 1]$ is said to be perfectly calibrated if, for any $b \in [0, 1]$, it holds that $P(Y = 1 \mid f_B(Z) = b) = b$. Unfortunately, using finite amounts of (calibration) data, one can only hope to construct approximately calibrated confidence functions. There exist many different notions of approximate calibration, which have been proposed over the years. Here, for concreteness, we adopt the notion of $\alpha$-calibration[5] introduced by Hébert-Johnson et al. [11], however, our theoretical results can be easily adapted to other notions of approximate calibration[6].

**Definition 2** (Calibration)**.** *A confidence function $f_B : \mathcal{Z} \rightarrow [0, 1]$ satisfies $\alpha$-calibration with respect to $\mathcal{S} \subseteq \mathcal{Z}$ if there exists some $\mathcal{S}' \subseteq \mathcal{S}$, with $|\mathcal{S}'| \geq (1 - \alpha)|\mathcal{S}|$, such that, for any $b \in [0, 1]$, it holds that*

$$|P(Y = 1 \mid f_B(Z) = b, Z \in \mathcal{S}') - b| \leq \alpha, \tag{5}$$

If the decision maker's decision $T$ only depends on the classifier's confidence $B$, *i.e.*, $\pi(H, B, W) = \pi(B)$ and $f_B$ satisfies $\alpha$-calibration with respect to $\mathcal{Z}$, then, it readily follows from previous work that, for any utility function that satisfies Eq. 1, a simple monotone AI-assisted decision policy $\pi_B^*$ that takes decisions by thresholding the confidence values is optimal [4–7], *i.e.*, $\pi_B^* = \text{argmax}_{\pi \in \Pi(B)} \mathbb{E}_\pi[u(T, Y)]$, where the expectation is with respect to the probability distribution $P^{\mathcal{M}}$ and $\Pi(B)$ denotes the class of AI-assisted decision policies using $B$. However, one of the main motivations to favor AI-assisted decision making over fully automated decision making is that the decision maker may have access to additional features $V$ and may like to weigh the classifier's confidence $B$ against her own confidence $H$. Hence, the decision maker may seek for the optimal decision policy $\pi^*$ over the class $\Pi(H, B)$ of AI-assisted decision policies using $H$ and $B$, *i.e.*, $\pi^* = \text{argmax}_{\pi \in \Pi(H,B)} \mathbb{E}_\pi[u(T, Y)]$, since it may offer greater expected utility than $\pi_B^*$.

Unfortunately, the following negative result shows that, in general, a rational decision maker may be unable to discover such an optimal decision policy $\pi^*$ using (perfectly) calibrated confidence values and this is true even if $f_H$ is monotone:

---

[5]Note that, if $\alpha = 0$ and $\mathcal{S} = \mathcal{Z}$, the confidence function $f$ is perfectly calibrated.

[6]All proofs can be found in Appendix A.

**Theorem 3.** *There exist (infinitely many) AI-assisted decision making processes $\mathcal{M}$ satisfying Eqs. 2 and 3, with utility functions $u(T, Y)$ satisfying Eq. 1, such that $f_B$ is perfectly calibrated and $f_H$ is monotone but any AI-assisted decision policy $\pi \in \Pi(H, B)$ that satisfies monotonicity is suboptimal,* i.e., $\mathbb{E}_\pi[u(T, Y)] < \mathbb{E}_{\pi^*}[u(T, Y)]$.

In the proof of the above result in Appendix A.2, we show that there always exist a perfectly calibrated $f_B(Z) = f_B(X, H)$ that depends on both $X$ and $H$ for which any monotone AI-assisted decision policy is suboptimal. This is due to the fact that $f_B(Z)$ is calibrated on average over $H$, however, it may not be calibrated, nor even monotone, after conditioning on a specific value $H = h$. Further, we also show that, even if $f_B(Z) = P^{\mathcal{M}}(Y = 1 \mid X)$ matches the true distribution of the label $Y$ given the features $X$, which has been typically the ultimate goal in the machine learning literature, there always exists a monotone $f_H$ for which any monotone AI-assisted decision policy is suboptimal. This is due to the fact that the decision maker's confidence $H$ can differ across instances with the same value for features $X$ because it also depends on the features $V$ and noise $Q$. Hence, $f_H$ may not be monotone after conditioning on a specific value $X = x$. In both cases, when a rational decision maker compares pairs of confidence values $h, b$ and $h', b'$, the rate of positive outcomes $Y = 1$ for each pair may appear *contradictory* with the magnitude of confidence. In what follows, we will show that, if $f_B$ satisfies a natural alignment property with respect to $f_H$, which we refer to as human-alignment, there always exists an optimal AI-assisted decision policy that is monotone.

## 4 AI-Assisted Decision Making Under Human-Aligned Calibration

Intuitively, to avoid that pairs of confidence values $B$ and $H$ appear as contradictory to a rational decision maker, we need to make sure that, with high probability, both $f_B$ and $f_H$ are monotone after conditioning on specific values of $H$ and $B$, respectively. Next, we formalize this intuition by means of the following property, which we refer to as $\alpha$-alignment:

**Definition 4** (Human-alignment). *A confidence function $f_B$ satisfies $\alpha$-alignment with respect to a confidence function $f_H$ if, for any $h \in \mathcal{H}$, there exists some $\tilde{\mathcal{S}}_h \subseteq \mathcal{S}_h$, with $\mathcal{S}_h = \{(x, H) \in \mathcal{Z} \mid H = h\}$ and $|\tilde{\mathcal{S}}_h| \geq (1 - \alpha/2)|\mathcal{S}_h|$, such that, for any $b', b'' \in [0, 1]$ and $h', h'' \in \mathcal{H}$ such that $b' \leq b''$ and $h' \leq h''$, it holds that*

$$P(Y = 1 \mid f_B(X, H) = b', (X, H) \in \tilde{\mathcal{S}}_{h'}) - P(Y = 1 \mid f_B(X, H) = b'', (X, H) \in \tilde{\mathcal{S}}_{h''}) \leq \alpha \tag{6}$$

The above definition just means that, if $f_B$ is $\alpha$-aligned with respect to $f_H$ then, for any $h, h' \in \mathcal{H}$, we can bound any violation of monotonicity by $f_B$ between at least a $(1 - \alpha/2)$ fraction of the subspaces of features $\mathcal{S}_h$ and $\mathcal{S}_{h'}$. Moreover, note that, if $f_B$ is 0-aligned with respect to $f_H$, then there are no violations of monotonicity, *i.e.*, $P(Y = 1 \mid f_B(X, H) = b', (X, H) \in \tilde{\mathcal{S}}_{h'}) \leq P(Y = 1 \mid f_B(X, H) = b'', (X, H) \in \tilde{\mathcal{S}}_{h''})$, and we say that $f_B$ is perfectly aligned with respect to $f_H$.

Given the above definition, we are now ready to state our main result, which shows that human-alignment allows for AI-assisted decision policies that satisfy monotonicity and (near-)optimality:

**Theorem 5.** *Let $\mathcal{M}$ be any AI-assisted decision making process satisfying Eqs. 2 and 3, with an utility function $u(T, Y)$ satisfying Eq. 1 If $f_B$ satisfies $\alpha$-alignment w.r.t. $f_H$, then there always exists an AI-assisted decision policy $\pi \in \Pi(H, B)$ that satisfies monotonicity and is near-optimal,* i.e.,

$$\mathbb{E}_{\pi^*}[u(T, Y)] \leq \mathbb{E}_\pi[u(T, Y)] + \alpha \cdot \left[ u(1, 1) - u(0, 1) + \frac{3}{2}(u(0, 0) - u(1, 0)) \right] \tag{7}$$

*where $\pi^* = \mathrm{argmax}_{\pi \in \Pi(H, B)} \mathbb{E}_\pi[u(T, Y)]$ is the optimal policy.*

**Corollary 1.** *If $f_B$ is perfectly aligned with respect to $f_H$, then there always exists an AI-assisted decision policy $\pi \in \Pi(H, B)$ that satisfies monotonicity and is optimal.*

Finally, in many high-stakes applications, we may like to make sure that the confidence values provided by $f_B$ are both useful and interpretable [34]. Hence, we may like to seek for confidence functions $f_B$ that satisfy human-aligned calibration, which we define as follows:

**Definition 6** (Human-aligned calibration). *A confidence function $f_B$ satisfies $\alpha$-aligned calibration with respect to a confidence function $f_H$ if and only if $f_B$ satisfies $\alpha$-alignment with respect to $f_H$ and it satisfies $\alpha$-calibration with respect to $\mathcal{Z}$.*

In the next section, we will show how to achieve human-alignment and human-aligned calibration via multicalibration, a statistical notion introduced in the context of algorithmic fairness [11].

# 5 Achieving Human-Aligned Calibration via Multicalibration

Multicalibration was introduced by Hébert-Johnson et al. [11] as a notion to achieve fairness in supervised learning. It strengthens the notion of calibration by requiring that the confidence function is calibrated simultaneously across a large collection of subspaces of features $\mathcal{C} \subseteq 2^{\mathcal{Z}}$ which may or may not be disjoint. More formally, it is defined as follows:

**Definition 7** (Multicalibration). *A confidence function $f_B : \mathcal{Z} \to \mathcal{B}$ satisfies $\alpha$-multicalibration with respect to $\mathcal{C} \subseteq 2^{\mathcal{Z}}$ if $f_B$ satisfies $\alpha$-calibration with respect to every $\mathcal{S} \in \mathcal{C}$.*

Then, we can show that, for an appropriate choice of $\mathcal{C}$, if $f_B$ satisfies $\alpha$-multicalibration with respect to $\mathcal{C}$, then it satisfies $\alpha$-aligned calibration with respect to $f_H$. More specifically, we have the following result:

**Theorem 8.** *If $f_B$ satisfies $(\alpha/2)$-multicalibration with respect to $\{\mathcal{S}_h\}_{h \in \mathcal{H}}$, with $\mathcal{S}_h = \{(x, H) \in \mathcal{Z} \mid H = h\}$, then $f_B$ satisfies $\alpha$-aligned calibration with respect to $f_H$.*

The above theorem suggests that, given a classifier's confidence function $f_B$, we can multicalibrate $f_B$ with respect to $\{\mathcal{S}_h\}_{h \in \mathcal{H}}$ to achieve $\alpha$-aligned calibration with respect to $f_H$. To achieve multicalibration guarantees using finite amounts of (calibration) data, multicalibration algorithms need to discretize the range of $f_B$ [9, 11, 12]. In what follows, we briefly revisit two algorithms, which carry out this discretization differently, and discuss their complexity and data requirements with respect to achieving $\alpha$-aligned calibration.

**Multicalibration algorithm via $\lambda$-discretization.** This algorithm, which was introduced by Hébert-Johnson et al. [11], discretizes the range of $f_B$, *i.e.*, the interval $[0, 1]$, into bins of fixed size $\lambda > 0$ with values $\Lambda[0, 1] = \{\frac{\lambda}{2}, \frac{3\lambda}{2}, \ldots, 1 - \frac{\lambda}{2}\}$.

Let $\lambda(b) = [b - \lambda/2, b + \lambda/2)$. The algorithm partitions each subspace $\mathcal{S}_h$ into $1/\lambda$ groups $\mathcal{S}_{h,\lambda(b)} = \{(x, h) \in \mathcal{S}_h \mid f_B(x, h) \in \lambda(b)\}$, with $b \in \Lambda[0, 1]$. It iteratively updates the confidence values of function $f_B$ for these groups until $f_B$ satisfies a discretized notion of $\alpha'$-multicalibration over these groups. The algorithm then returns a discretized confidence function $f_{B,\lambda}(x, h) = \mathbb{E}[f_B(X, H) \mid f_B(X, H) \in \lambda(b)]$, with $b \in \Lambda[0, 1]$ such that $f_B(x, h) \in \lambda(b)$, which is guaranteed to satisfy $(\alpha' + \lambda)$-multicalibration. Refer to Algorithm 1 in Appendix B for a pseudocode of the algorithm.

Then, as a direct consequence of Theorem 8, we can obtain a (discretized) confidence function $f_{B,\lambda}$ that satisfies $\alpha$-aligned calibration by setting $\alpha' = \lambda = \alpha/4$. However, the following proposition shows that, to satisfy just $\alpha$-alignment, it is enough to set $\alpha' = \frac{3}{8}\alpha > \alpha/4$ and $\lambda = \alpha/4$:

**Proposition 1.** *The discretized confidence function $f_{B,\lambda}$ returned by Algorithm 1 satisfies $(2\alpha' + \lambda)$-alignment with respect to $f_H$.*

Finally, it is worth noting that, to implement Algorithm 1, we need to compute empirical estimates of the expectations and probabilities above using a calibration set $\mathcal{D}$. In this context, Theorem 2 in Hébert-Johnson et al. [11] shows that, if we use a calibration set of size $O(\log(|\mathcal{H}|/(\alpha\gamma\xi))/\alpha^{11/2}\gamma^{3/2})$, with $P((X, H) \in \mathcal{S}_h) > \gamma$ for all $h \in \mathcal{H}$, then $f_{B,\lambda}$ is guaranteed to satisfy $\alpha$-multicalibration with probability at least $1 - \xi$ in time $O(|\mathcal{H}| \cdot \text{poly}(1/\alpha, 1/\gamma))$.

**Multicalibration algorithm via uniform mass binning.** Uniform mass binning (UMD) [9, 12] has been originally designed to calibrate $f_B$ with respect to $\mathcal{Z}$ using a calibration set $\mathcal{D}$. However, since the subspaces $\{\mathcal{S}_h\}_{h \in \mathcal{H}}$ are disjoint, *i.e.*, $\mathcal{S}_h \cap \mathcal{S}_{h'} = \emptyset$ for every $h \neq h'$, we can multicalibrate $f_B$ with respect to $\{\mathcal{S}_h\}_{h \in \mathcal{H}}$ by just running $|\mathcal{H}|$ instances of UMD, each using the subset of samples $\mathcal{D} \cap \mathcal{S}_h$. Here, we would like to emphasize that we can use UMD to achieve multicalibration because, in our setting, the subspaces $\{\mathcal{S}_h\}_{h \in \mathcal{H}}$ are disjoint.

Each instance of UMD discretizes the range of $f_B$, *i.e.*, the interval $[0, 1]$, into $N = 1/\lambda$ bins with values $\Lambda_h[0, 1] = \{\hat{P}(Y = 1 \mid f_B(X, h) \in [0, \hat{q}_1]), \ldots, \hat{P}(Y = 1 \mid f_B(X, h) \in [\hat{q}_{N-1}, \hat{q}_N])\}$, where $\hat{q}_i$ denotes the $(i/N)$-th empirical quantile of the confidence values $f_B(x, h)$ of the samples $(x, h) \in \mathcal{D} \cap \mathcal{S}_h$ and $\hat{P}$ denotes an empirical estimate of the probability using samples from $\mathcal{D} \cap \mathcal{S}_h$, aswell. Here, note that, by construction, the bins have similar probability mass. Then, for each $(x, h) \in \mathcal{Z}$, the corresponding instance of UMD provides the value of the discretized confidence function $f_{B,\lambda}(x, h) = b$, where $b \in \Lambda_h[0, 1]$ denotes the value of the bin whose corresponding defining interval includes $f_B(x, h)$. Finally, we have the following theorem, which guarantees that, as

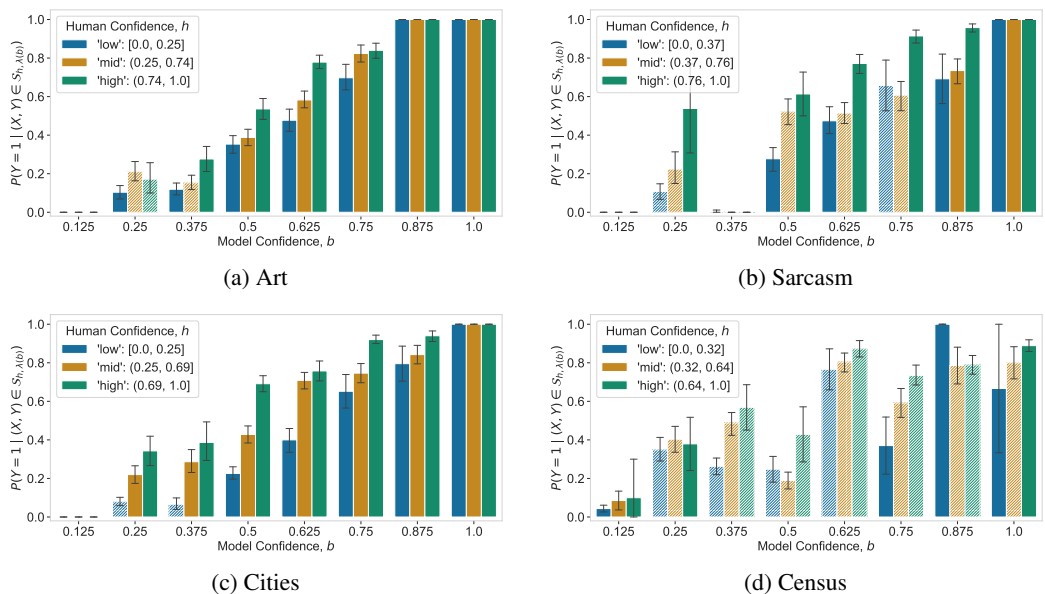

Figure 2: Empirical estimate of the probabilities $P(Y = 1 \mid (X, Y) \in \mathcal{S}_{h,\lambda(b)})$, where $b \in \Lambda[0, 1]$ and $h \in \{\text{low}, \text{mid}, \text{high}\}$ are the discretized confidence values for the classifiers and human participants, respectively. Error bars represent 90% confidence intervals and hatched bars mark alignment violations between confidence pairs $(h, b)$ with $|\mathcal{S}_{h,\lambda(b)}| \geq 30$.

long as the calibration set is large enough, the discretized confidence function $f_{B,\lambda}$ satisfies $\alpha$-aligned calibration with respect to $f_H$ with high probability:

**Theorem 9.** *The discretized confidence function $f_{B,\lambda}$ returned by $|\mathcal{H}|$ instances of UMD, one per $\mathcal{S}_h$, satisfies $\alpha$-aligned calibration with respect to $f_H$ with probability at least $1 - \xi$ as long as the size of the calibration set $|\mathcal{D}| = O\left(\frac{|\mathcal{H}|}{\alpha^2 \lambda \gamma} \log\left(\frac{|\mathcal{H}|}{\lambda \xi}\right)\right)$, with $P((X, H) \in \mathcal{S}_h) \geq \gamma$.*

## 6 Experiments

In this section, we validate our theoretical results using a dataset with real expert predictions in an AI-assisted decision making scenario comprising four different binary classification tasks[7].

**Data description.** We experiment with the publicly available Human-AI Interactions dataset [35]. The dataset comprises 34,783 unique predictions from 1,088 different human participants on four different binary prediction tasks ("Art", "Sarcasm", "Cities" and "Census"). Overall, there are approximately 32 different instances per task. In the "Art" task, participants need to determine the art period of a painting given two choices and, overall, there are paintings from four art periods. In the "Sarcasm" task, participants need to detect if sarcasm is present in text snippets from the Reddit sarcasm dataset [36]. In the "Cities" task, participants need to determine which large US city is depicted in an image given two choices and, overall, there are images of four different US cities. Finally, in the "Census" task, participants need to determine if an individual earns more than 50k a year based on certain demographic information in tabular form. For "Sarcasm", $x$ is a representation of the text snippets and we set $y = 1$ if sarcasm is present, for "Art" and "Cities", $x$ is a representation of the images and we set $y = 1$ and $y = 0$ at random for each different instance and, for "Census", $x$ summarizes demographic information and we set $y = 1$ if an individual earns more than 50k a year. In each of the tasks, human participants provide confidence values about their predictions before ($h$) and after ($h_{+\text{AI}}$) receiving AI advice from a classifier in form of the classifier's confidence values $b$.[8] The original dataset contains predictions by participants from different, but overlapping, sets

---

[7]We release the code to reproduce our analysis at https://github.com/Networks-Learning/human-aligned-calibration.

[8]Refer to Appendix C for more details on the dataset.

Table 1: Misalignment, miscalibration and AUC.

| Task | Misalignment | | Miscalibration | | AUC | | |
|---|---|---|---|---|---|---|---|
| | EAE | MAE | ECE | MCE | $\pi_B$ | $\pi_H$ | $\pi_{H_{+AI}}$ |
| Art | $4.5 \cdot 10^{-4}$ | 0.058 | 0.084 | 0.186 | 86.7% | 72.7% | 82.0% |
| Sarcasm | $3.8 \cdot 10^{-3}$ | 0.224 | 0.085 | 0.310 | 89.9% | 82.5% | 86.5% |
| Cities | $6.2 \cdot 10^{-5}$ | 0.013 | 0.066 | 0.158 | 84.4% | 79.0% | 84.7% |
| Census | $9.0 \cdot 10^{-3}$ | 0.298 | 0.109 | 0.270 | 80.0% | 77.3% | 79.9% |

of countries across tasks, who were told the AI advice had different values of accuracy.[9] In our experiments, to control for these confounding factors, we focus on participants from the US who were told the AI advice was $80\%$ accurate, resulting in $15,063$ unique predictions from $471$ different human participants.

**Experimental setup and evaluation metrics.** For each of the tasks, we first measure (i) the degree of misalignment between the classifiers' confidence values $b$ and the participants' confidence values $h$ before receiving AI advice $b$ and (ii) the difference $(h_{+AI} - h)$ between the human participant's confidence values before and after receiving AI advice $b$. Then, we compare the utility achieved by a AI-assisted decision policy $\pi_{H_{+AI}}$ that predicts the value of $y$ by thresholding the humans' confidence values $h_{+AI}$ after observing the classifier's confidence values against two baselines: (i) a decision policy $\pi_B$ that predicts the value of $y$ by thresholding the classifier's confidence values $b$ and (ii) a decision policy $\pi_H$ that predicts the value of $y$ by thresholding the humans' confidence values $h$ before observing the classifier's confidence values.

To measure the degree of misalignment, we discretize the confidence values $b$ and $h$ into bins. For the classifiers' confidence $b$, we use $8$ uniform sized bins per task with (centered) values $\Lambda[0, 1]$, where $\lambda = 1/8$. For the human participants' confidence $h$ before receiving AI advice $b$, we use three bins per task ('low', 'mid' and 'high'), where we set the bin boundaries so that each bin contains approximately the same probability mass and set the bin values to the average confidence value within each bin. In what follows, we refer to the pairs of discretized confidence values $(h, b)$ as cells, where samples $(x, y) \in \mathcal{Z}$ whose confidence values lie in the cell $(h, b)$ define the group $\mathcal{S}_{h,\lambda(b)}$, and note that we choose a rather low number of bins for both $b$ and $h$ so that most cells have sufficient data samples to reliable estimate several misalignment metrics, which we describe next.

We use three different misalignment metrics: (i) the number of alignment violations between cell pairs, (ii) the expected alignment error (EAE) and (iii) the maximum alignment error (MAE). There is an alignment violation between cells pairs $(h, b)$ and $(h', b')$, with $h \leq h'$ and $b \leq b'$, if

$$P(Y = 1 | (X, Y) \in S_{h,\lambda(b)}) > P(Y = 1 | (X, Y) \in S_{h',\lambda(b')}).$$

Moreover, we have that:

$$\text{EAE} = \frac{1}{N} \cdot \sum_{h \leq h', b \leq b'} \left[ P(Y = 1 \mid (X, Y) \in \mathcal{S}_{h,\lambda(b)}) - P(Y = 1 \mid (X, Y) \in \mathcal{S}_{h',\lambda(b')}) \right]_+,$$

$$\text{MAE} = \max_{h \leq h', b \leq b'} P(Y = 1 \mid (X, Y) \in \mathcal{S}_{h,\lambda(b)}) - P(Y = 1 \mid (X, Y) \in \mathcal{S}_{h',\lambda(b')}),$$

where $N = |\{h \leq h', b \leq b'\}|$. Here, note that the number of alignment violations tells us how frequently is the left hand side of Eq. 6 positive across cell pairs given $\tilde{S}_h = S_h$ and the EAE and MAE quantify the average and maximum value of the left hand side of Eq. 6 across cells violating alignment. To obtain reliable estimates of the above metrics, we only consider cells $(h, b)$ with $|\mathcal{S}_{h,\lambda(b)}| \geq 30$ samples. Moreover, we also report the expected calibration error (ECE) and maximum calibration error (MCE) [12, 37], which are natural counterparts to EAE and MAE, respectively.

As a measure of utility, we estimate the true positive rate (TPR) and false positive rate (FPR) of the decision policies $\pi_B$, $\pi_H$ and $\pi_{H_{+AI}}$ for all possible choices of threshold values, which we summarize using the area under the ROC curve (AUC) and, in Appendix C, we also report ROC curves.

---

[9]Participants were also either told that the advice is from a "Human" or from an "AI" based on a random assignment of participants to a treatment or control group. Since the actual advice received in both groups was identical for the same instance and the "perceived advice source" is randomized, we use data from both treatment and control groups in the experiments.

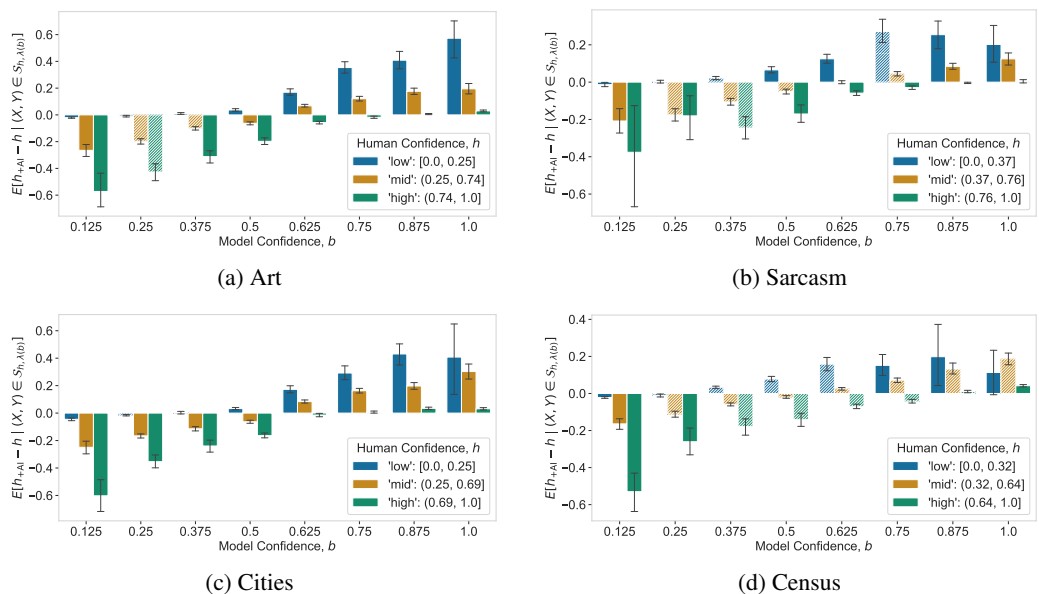


(a) Art        (b) Sarcasm



(c) Cities        (d) Census


Figure 3: Empirical estimate of the average difference $\mathbb{E}[h_{+\text{AI}} - h \mid (X, Y) \in \mathcal{S}_{h,\lambda(b)}]$, where $b \in \Lambda[0,1]$ and $h \in \{\text{low}, \text{mid}, \text{high}\}$ are the discretized confidence values for the classifier and human participants, respectively. Error bars represent 90% confidence intervals and hatched bars mark alignment violations between confidence pairs $(h, b)$ with $|\mathcal{S}_{h,\lambda(b)}| \geq 30$.

**Results.** We start by looking at the empirical estimates of the probabilities $P(Y = 1 \mid (X, Y) \in \mathcal{S}_{h,\lambda(b)})$ and of our measures of misalignment (EAE, MAE) and miscalibration (ECE, MCE) in Figure 2 and Table 1 (left and middle columns). The results show that, for "Cities", the probabilities $P(Y = 1 \mid (X, Y) \in \mathcal{S}_{h,\lambda(b)})$ are (approximately) monotonically increasing with respect to the classifier's confidence values $b$. More specifically, as shown in Figure 2, there is only one alignment violation between cell pairs and, hence, our metrics of misalignment acquire also very low values. In contrast, for "Art", "Sarcasm" and especially "Census", there is an increasing number of alignment violations and our misalignment metrics acquire higher values, up to several orders of magnitude higher for "Census". These results also show that misalignment and miscalibration go hand in hand, however, in terms of miscalibration, "Census" does not stand up so strongly.

Next, we look at the difference $h_{+\text{AI}} - h$ between the human participant's recorded confidence values before and after receiving AI advice $b$ across samples in each of the subsets $\mathcal{S}_{h,\lambda(b)}$ induced by the discretized confidence values used above. Figure 3 summarizes the results, which reveal that the difference $h_{+\text{AI}} - h$ increases monotonically with respect to the classifier's confidence $b$. This suggests that participants always expect $b$ to reflect the probability of a positive outcome irrespectively of their confidence value $h$ before receiving AI advice, providing support for our hypothesis that (rational) decision makers implement monotone AI-assisted decisions policies. Further, this finding also implies that, for "Art", "Sarcasm" and "Census", any policy $\pi_{H_{+\text{AI}}}$ that predicts the value of the label $y$ by thresholding the confidence value $h_{+\text{AI}}$ will be necessarily suboptimal because the probabilities $P(Y = 1 \mid (X, Y) \in \mathcal{S}_{h,\lambda(b)})$ are not monotone increasing with $b$.

Finally, we look at the AUC achieved by decision policies $\pi_B$, $\pi_H$ and $\pi_{H_{+\text{AI}}}$. Table 1 (right columns) summarize the results, which shows that $\pi_{H_{+\text{AI}}}$ outperforms $\pi_H$ consistently across all tasks but it only outperforms $\pi_B$ in a single task ("Cities") out of four. These findings provide empirical support for Theorem 3, which predicts that, in the presence of human-alignment violations as those observed in "Art", "Sarcasm" and "Census", any monotone AI-assisted decision policy will be suboptimal, and they also provide support for Theorem 5, which predicts that, under human-alignment, there exist near-optimal AI-assisted decision policies satisfying monotonicity.

## 7   Discussion and Limitations

In this section, we discuss the intended scope of our work and identify several limitations of our theoretical and experimental results, which may serve as starting points for future work.

**Decision making setting.** We have focused on decision making settings where both decisions and outcomes are binary. However, we think that it may be feasible to extend our theoretical analysis to settings with multi-categorical (or real-valued) outcome variables and decisions. One of the main challenges would be to identify which natural conditions utility functions may satisfy in such settings. Further, we also think that it would be significantly more challenging to extend our theoretical analysis to sequential settings—multicalibration in sequential settings is an open area of research—but our ideas may still be a useful starting point. In addition, our theoretical analysis assumes that the decision makers aim to maximize the average utility of their decisions. However, whenever human decisions are consequential to individuals, the decision maker may have fairness desiderata.

**Confidence values.** In our causal model of AI-assisted decision making, we allow the classifier's confidence values to depend on the decision maker's confidence values because this is necessary to achieve human-alignment via multicalibration as described in Section 5. However, we would like to clarify that both Theorems 3 and 5 still hold if the classifier's confidence values do not depend on the decision maker's confidence, as it is typically the status quo today. Looking into the future, our work questions this status quo by showing that, by allowing the classifier's confidence values to depend on the decision maker's confidence values, a decision maker may end up taking decisions with higher utility. Moreover, we would also like to clarify that, while the motivation behind our work is AI-assisted human decision making, our theoretical results do not depend on who—be it a classifier or another human—gives advice. As long as the advice comes in the form of confidence values, our results are valid. Finally, while we have shown that human-alignment can be achieved via multicalibration, we hypothesize that algorithms specifically designed to achieve human-alignment may have lower data and computational requirements than multicalibration algorithms.

**Experimental results.** Our experimental results demonstrate that, *across tasks*, the average utility achieved by decision makers is relatively higher if the classifier they use satisfies human-alignment. However, they do not empirically demonstrate that, *for a fixed task*, there is an improvement in average utility achieved by decision makers if the classifier they use satisfies human-alignment. The reason why we could not demonstrate the latter is because, in our experiments, we used an observational dataset gathered by others [35]. Looking into the future, it would be very important to run a human subject study to empirically demonstrate the latter and, for now, treat our conclusions with caution.

## 8 Conclusions

We have introduced a theoretical framework to investigate what properties confidence values should have to help decision makers take better decisions. We have shown that there exists data distribution for which a rational decision maker using calibrated confidence values will always take suboptimal decisions. However, we have further shown that, if the confidence values satisfy a natural alignment property, which can be achieved via multicalibration, then a rational decision maker using these confidence values can take optimal decisions. Finally, we have illustrated our theoretical results using real human predictions on four AI-assisted decision making tasks.

**Acknowledgements.** We would like to thank Nastaran Okati for fruitful discussions at an early stage of the project. Gomez-Rodriguez acknowledges support from the European Research Council (ERC) under the European Union's Horizon 2020 research and innovation programme (grant agreement No. 945719).

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

# A   Proofs

## A.1   Additional Lemmas

**Lemma 1** (Monotonicity). *If a utility function $u$ satisfies Eq. 1, then $u$ is monotone with respect to the probability that $Y = 1$, i.e., for any $P, P' \in \mathcal{P}(\{0, 1\})$ such that $P(Y = 1) \leq P'(Y = 1)$, it holds that $\mathbb{E}_{Y \sim P}[u(1, Y)] \leq \mathbb{E}_{Y \sim P'}[u(1, Y)]$.*

*Proof.* We readily have that
$$\mathbb{E}_{Y \sim P}[u(1, Y)] = P(Y = 1) \cdot u(1, 1) + (1 - P(Y = 1)) \cdot u(1, 0)$$
$$\leq P'(Y = 1) \cdot u(1, 1) + (1 - P'(Y = 1)) \cdot u(1, 0)$$
$$= \mathbb{E}_{Y \sim P'}[u(1, Y)],$$
where, in the above inequality, we use that $u(1, 1) > u(1, 0)$ and $P(Y = 1) \leq P'(Y = 1)$.   $\square$

**Lemma 2** (Trivial policies are not always optimal). *If a utility function $u$ satisfies Eq. 1, then there exist $P, P' \in \mathcal{P}(\{0, 1\})$ such that the trivial policies $\pi$ that either always decide $T = 1$ or always decide $T = 0$ are suboptimal. In particular, for any $P, P' \in \mathcal{P}(\{0, 1\})$ such that $P(Y = 1) < c$ and $P'(Y = 1) > c$, where*

$$c = \frac{u(0, 0) - u(1, 0)}{u(1, 1) - u(1, 0) + u(0, 0) - u(0, 1)} \in (0, 1), \tag{8}$$

*it holds that*

$$\mathbb{E}_{Y \sim P}[u(1, Y)] < \mathbb{E}_{Y \sim P}[u(0, Y)] \quad and \quad \mathbb{E}_{Y \sim P'}[u(1, Y)] > \mathbb{E}_{Y \sim P'}[u(0, Y)]. \tag{9}$$

*Proof.* Let $P$ be any distribution such that
$$P(Y = 1) < c = \frac{u(0, 0) - u(1, 0)}{u(1, 1) - u(1, 0) + u(0, 0) - u(0, 1)},$$
where $c \in (0, 1)$ because, by assumption, $u$ satisfies Eq. 1. Now, by rearranging the above inequality, we have that
$$P(Y = 1) \cdot u(1, 1) + (1 - P(Y = 1)) \cdot u(1, 0) < P(Y = 1) \cdot u(0, 1) + (1 - P(Y = 1)) \cdot u(0, 0),$$
and, using the definition of the expectation, it immediately follows that
$$\mathbb{E}_{Y \sim P}[u(1, Y)] < \mathbb{E}_{Y \sim P}[u(0, Y)].$$
The same argument can be used to show that, for any distribution $P'$ such that $P'(Y = 1) > c$, it holds that $\mathbb{E}_{Y \sim P'}[u(1, Y)] > \mathbb{E}_{Y \sim P'}[u(0, Y)]$. Finally, note that, since $c \in (0, 1)$, we know that such distributions $P$ and $P'$ exist.   $\square$

## A.2   Proof of Theorem 3

Before proving Theorem 3, we rewrite the expected utility with respect to the probability distribution $P^{\mathcal{M}}$ in terms of confidence $H$ and $B$ by using the law of total expectation,
$$\mathbb{E}_{\pi}[u(T, Y)] = \mathbb{E}_{H, B \sim P^{\mathcal{M}}(H, B)}\left[\mathbb{E}_{\pi}[u(T, Y) | H, B]\right].$$
Here, to simplify notation, we will write
$$\mathbb{E}_{H, B}\left[\mathbb{E}_{\pi}[u(T, Y) \mid H, B]\right],$$
where note that, using the law of total expectation, we can write the inner expectation in the above expression in terms of the utilities of the trivial policies, *i.e.,*
$$\mathbb{E}_{\pi}[u(T, Y) \mid H, B] = \mathbb{E}[u(1, Y) \mid H, B] \cdot P_{\pi}(T = 1 \mid H, B)$$
$$+ \mathbb{E}[u(0, Y) \mid H, B] \cdot P_{\pi}(T = 0 \mid H, B), \tag{10}$$
and we will use $P$ to refer to probabilities induced by SCM $\mathcal{M}$, *e.g.,* $P(H, B)$ to denote $P^{\mathcal{M}}(H, B)$. Now, we restate and prove Theorem 3.

**Theorem 3.** There exist (infinitely many) AI-assisted decision making processes $\mathcal{M}$ satisfying Eqs. 2 and 3, with utility functions $u(T, Y)$ satisfying Eq. 1, such that $f_B$ is perfectly calibrated and $f_H$ is monotone but any AI-assisted decision policy $\pi \in \Pi(H, B)$ that satisfies monotonicity is suboptimal, *i.e.,* $\mathbb{E}_{\pi}[u(T, Y)] < \mathbb{E}_{\pi^*}[u(T, Y)]$.

*Proof.* To prove the above claim, we construct a monotone confidence function $f_H$, perfectly calibrated confidence function $f_B$ and distribution $P^{\mathcal{M}}$ for which any monotone AI-assisted decision policy $\pi \in \Pi(H, B)$ achieves strictly lower utility than a carefully constructed non monotone AI-assisted decision policy $\tilde{\pi} \in \Pi(H, B)$.

We will present the proof in three parts. First, we will introduce the main building block and idea behind the proof by a small construction of $f_H$, $f_B$ and $P^{\mathcal{M}}$ with $|\mathcal{H}| = |\mathcal{B}| = 3$, where $\mathcal{B} \subseteq [0, 1]$ denotes the (discrete) output space of the classifier's confidence function. We then construct examples of $f_H$, $f_B$ and $P^{\mathcal{M}}$ for arbitrary $|\mathcal{H}| = k$ and $|\mathcal{B}| = m$ with $m, k \in \mathbb{N}$, $m > k \geq 2$. Lastly, we construct examples where $\mathcal{B}$ is non-discrete and $|\mathcal{H}| = k$ with $k > 2$.

**Main building block and small example.**

We start by presenting the main idea of the proof using an example with a small set of confidence values $\mathcal{H}$ and $\mathcal{B}$. Let the values of the decision maker's confidence $H$ be in $\mathcal{H} = \{h_1, h_2, h_3\}$ and the values of the classifier's confidence $B$ be in $\mathcal{B} = \{b_1, b_2, b_3\}$, with order $h_i < (h_i + 1)$ and $b_i < (b_i + 1)$ respectively.

Our main building block, consists of two distributions $P^-, P^+ \in \mathcal{P}(\{0, 1\})$ with $P^-(Y = 1) < c$ and $P^+(Y = 1) > c$, where $c$ depends on utility $u$ as described by Eq. 8 in Lemma 2. We use these distributions for our constructions of $f_H$, $f_B$ and $P^{\mathcal{M}}$, so that for some realizations of $H, B$ distribution $P(Y = 1 \mid H, B)$ is either $P^-$ or $P^+$. Using Lemma 2 and from Eq. 10, we have that:

(I) For any $h_i, b_i$ such that $P(Y \mid H = h_i, B = b_i) = P^-$, it holds that
$$\mathbb{E}[u(1, Y) \mid H = h_i, B = b_i] < \mathbb{E}[u(0, Y) \mid H = h_i, B = b_i].$$
Hence, *decreasing* $P_\pi(T = 1 \mid H, B)$ *increases* $\mathbb{E}[u(T, Y) \mid H = h_i, B = b_i]$.

(II) For any $h_i, b_i$ such that $P(Y \mid H = h_i, B = b_i) = P^+$, it holds that
$$\mathbb{E}[u(1, Y) \mid H = h_i, B = b_i] > \mathbb{E}[u(0, Y) \mid H = h_i, B = b_i].$$
Hence, *increasing* $P_\pi(T = 1 \mid H, B)$ *increases* $\mathbb{E}[u(T, Y) \mid H = h_i, B = b_i]$.

Intuitively, suppose we now have that, for confidence values $h_2, b_2, Y \sim P^+$ and, for confidence values $h_3, b_2, Y \sim P^-$, *i.e.*, $P(Y \mid H = h_2, B = b_2) = P^+$ and $P(Y \mid H = h_3, B = b_2) = P^-$. Then, any non-monotone AI-assisted decision policy $\tilde{\pi}$ with $P_{\tilde{\pi}}(T = 1 \mid H = h_2, B = b_2) > P_{\tilde{\pi}}(T = 1 \mid H = h_3, B = b_2)$ will have higher expected utility than any monotone AI-assisted decision policy given confidence values $h_2, b_2$ and $h_3, b_2$. Finally, under an appropriate choice of distribution $P(H, B)$, such non-monotone AI-assisted decision policies $\tilde{\pi}$ will offer higher overall utility in expectation.

We formalize this intuition with the following lemma:

**Lemma 3.** *Let $\mathcal{M}$ be any AI-assisted decision making process satisfying Eqs. 2 and 3, with utility function $u(T, Y)$ satisfying Eq. 1. If $f_H$, $f_B$ and $P^{\mathcal{M}}$ are such that there exists confidence values $b \in \mathcal{B}$, $h_i, h_j \in \mathcal{H}$, with $h_i < h_j$, which satisfy*

$$P(H = h_i, B = b) > 0, \quad P(H = h_j, B = b) > 0,$$
$$P(Y \mid H = h_i, B = b) = P^+ \quad and \quad P(Y \mid H = h_j, B = b) = P^-, \tag{11}$$

*for some distributions $P^-, P^+$ with $P^-(Y = 1) < c$ and $P^+(Y = 1) > c$, where*

$$c = \frac{u(0, 0) - u(1, 0)}{u(1, 1) - u(1, 0) + u(0, 0) - u(0, 1)}. \tag{12}$$

*Then, for any monotone AI-assisted decision policy $\pi \in \Pi(H, B)$, there exists an AI-assisted decision policy $\tilde{\pi} \in \Pi(H, B)$ which is not monotone and achieves a stricly greater utility than $\pi$, i.e., $\mathbb{E}_\pi[u(T, Y)] < \mathbb{E}_{\tilde{\pi}}[u(T, Y)]$.*

*Proof.* Let $\pi$ be a monotone AI-assisted decision policy, then it must hold that $P_\pi(T = 1 \mid H = h_i, B = b) \leq P_\pi(T = 1 \mid H = h_j, B = b)$ (see Eq. 4). Let $\tilde{\pi}$ be an identical AI-assisted decision

policy to $\pi$ up to the decision for confidence values $h_i, b$ and $h_j, b$. We distinguish between three cases.

— **Case 1**: $P_\pi(T = 1 \mid H = h_i, B = b) < P_\pi(T = 1 \mid H = h_j, B = b)$.

Let the probability of $T = 1$ under $\tilde{\pi}$ for confidence values $h_i, b$ and $h_j, b$ be switched compared to $\pi$, *i.e.*,

$$P_{\tilde{\pi}}(T = 1 \mid H = h_i, B = b) = P_\pi(T = 1 \mid H = h_j, B = b),$$
$$P_{\tilde{\pi}}(T = 1 \mid H = h_j, B = b) = P_\pi(T = 1 \mid H = h_i, B = b).$$

Then, $\tilde{\pi}$ is not monotone, as Eq. 4 is not satisfied, and it holds that

$$P_{\tilde{\pi}}(T = 1 \mid H = h_i, B = b) > P_\pi(T = 1 \mid H = h_i, B = b),$$
$$P_{\tilde{\pi}}(T = 1 \mid H = h_j, B = b) < P_\pi(T = 1 \mid H = h_j, B = b).$$

As we decreased $P(T = 1 \mid H = h_j, B = b)$ and increased $P(T = 1 \mid H = h_i, B = b)$, by properties (I) and (II), it must hold that the expected utility of $\tilde{\pi}$ given confidence values $h_i, b$ and $h_j, b$ is higher than the one of $\pi$, *i.e.*,

$$\mathbb{E}_{\tilde{\pi}}[u(T, Y) \mid H = h_i, B = b] > \mathbb{E}_\pi[u(T, Y) \mid H = h_i, B = b] \quad \text{and} \tag{13}$$
$$\mathbb{E}_{\tilde{\pi}}[u(T, Y) \mid H = h_j, B = b] > \mathbb{E}_\pi[u(T, Y) \mid H = h_j, B = b]. \tag{14}$$

— **Case 2**: $0 < P_\pi(T = 1 \mid H = h_i, B = b) = P_\pi(T = 1 \mid H = h_j, B = b) \leq 1$.

Let the probability of $T = 1$ under $\tilde{\pi}$ for confidence values $h_j, b$ be strictly lower compared to $\pi$ and be the same as $\pi$ for $h_i, b$. Then, $\tilde{\pi}$ is not monotone, since by case assumption

$$P_{\tilde{\pi}}(T = 1 \mid H = h_i, B = b) = P_\pi(T = 1 \mid H = h_j, B = b) > P_{\tilde{\pi}}(T = 1 \mid H = h_j, B = b)$$

and the inequality in Eq. 14 holds by property (I).

— **Case 3**: $P_\pi(T = 1 \mid H = h_i, B = b) = P_\pi(T = 1 \mid H = h_j, B = b) = 0$.

Let the probability of $T = 1$ under $\tilde{\pi}$ for confidence values $h_i, b$ be strictly higher compared to $\pi$ and be the same as $\pi$ for $h_j, b$. Then, $\tilde{\pi}$ is not monotone, since by case assumption

$$P_{\tilde{\pi}}(T = 1 \mid H = h_j, B = b) = P_\pi(T = 1 \mid H = h_i, B = b) < P_{\tilde{\pi}}(T = 1 \mid H = h_i, B = b)$$

and the inequality in Eq. 13 holds by property (II).

As in all three cases at least one of the strict inequalities in Eqs. 13 or 14 holds and $\tilde{\pi}$ is equivalent to $\pi$ (*i.e.*, it has the same expected conditional utility) given any other pair of confidence values $h' \in \mathcal{H}$, $b' \in \mathcal{B}$, we have that

$$\mathbb{E}_{\tilde{\pi}}[u(T, Y)] = \mathbb{E}[\mathbb{E}_{\tilde{\pi}}[u(T, Y)] \mid H, B] > \mathbb{E}[\mathbb{E}_\pi[u(T, Y) \mid H, B] = \mathbb{E}_\pi[u(T, Y)].$$

$\square$

Before proceeding further, we would like to note that we may also state Lemma 3 using $h \in \mathcal{H}$, $b_i, b_j \in \mathcal{B}$, with $b_i < b_j$, the proof would follow analogously.

Now, we construct an AI-decision making process $\mathcal{M}$, with $\mathcal{H} = \{h_1, h_2, h_3\}$ and $\mathcal{B} = \{b_1, b_2, b_3\}$, such the decision maker's confidence $f_H$ is monotone, the classifier's confidence $f_B$ is perfectly calibrated, and the conditions of Lemma 3 are satisfied. First, let $f_H, f_B$ and $P^\mathcal{M}$ be such that

$$P(f_B(Z) = b_j) = \begin{cases} 3/6 & \text{if } j = 1 \\ 2/6 & \text{if } j = 2 \\ 1/6 & \text{if } j = 3 \\ 0 & \text{otherwise} \end{cases} \quad \text{and}$$

$$P(H = h_i \mid B = b_j) := P_{X,V}(H = h_i \mid f_B(Z) = b_j) = \begin{cases} \frac{1}{4-j} & \text{if } i \geq j \\ 0 & \text{otherwise.} \end{cases}$$

Then, it readily follows that $P(H = h_i, B = b_j) = 1/6$ for $i \geq j$ and $P(H = h_i, B = b_j) = 0$ otherwise. Moreover, for each pair of confidence values $(h_i, b_j)$ with positive probability $P(H = h_i, B = b_j)$, we set

$$P(Y = 1 \mid H = h_i, B = b_j) = \begin{cases} P^+ & \text{if } i = j = 2 \text{ or } (i = 3 \text{ and } j \in \{1, 3\}) \\ P^- & \text{if } (j = 2 \text{ and } i = 3) \text{ or } (j = 1 \text{ and } i \in \{1, 2\}), \end{cases}$$

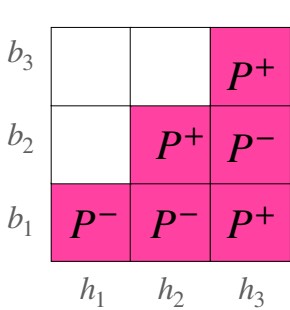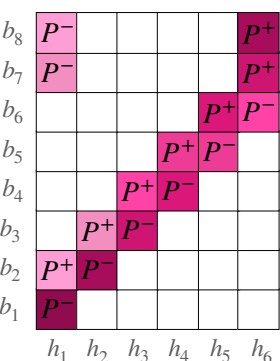

Figure 4: Nonzero values of $P(Y = 1|H = h_i, B = b_j)$ and $P(H = h_i, B = b_j)$ for every $h_i \in \mathcal{H}$ and $b_j \in \mathcal{B}$ used in the first (left) and second (right) part of the proof of Theorem 3. In each cell $(h_i, b_j)$ in both panels, $P^+$ or $P^-$ is the value of $P(Y = 1|H = h_i, B = b_j)$ and lighter color means lower value of $P(H = h_i, B = b_j)$, where white means $P(Y = 1|h = h_i, B = b_j) = 0$ and $P(H, B) = 0$. In both panels, the assignment of values is very stylized to facilitate the proof—the classifier's confidence function $f_B$ partitions the feature space in a way such that a rational decision maker is unable to take decisions that maximize utility for almost all confidence values. However, less stylized examples also satisfy the conditions of Lemma 3. For example, as long as there is one triplet of confidence values $b_2, h_2, h_3$ (or $h_3, b_1, b_2$ in the left example) for which a rational decision maker is unable to take decisions that maximize utility, Lemma 3 can be applied.

as shown in Figure 4 (left). Then, it readily follows that $f_H$ is monotone with respect to the probability that $Y = 1$, *i.e.*, $P(Y = 1 \mid H = h_i) \leq P(Y = 1 \mid H = h_{i+1})$), and we have that the classifier's confidence values

$$b_j := \sum_{i:i \geq j} P(H = h_i \mid B = b_j) \cdot P(Y = 1 \mid H = h_i, B = b_j)$$

$$= \begin{cases} 2/3 \cdot P^- + 1/3 \cdot P^+ & \text{if } j = 1 \\ 1/2 \cdot P^- + 1/2 \cdot P^+ & \text{if } j = 2 \\ P^+ & \text{if } j = 3 \\ 0 & \text{otherwise} \end{cases}$$

are perfectly calibrated and satisfy that $b_j < b_{j+1}$.

Finally, using Lemma 3 with $b = b_2$, $h_i = h_2$, $h_j = h_3$, we have that any monotone AI-assisted decision policy is suboptimal for any $\mathcal{M}$ with $f_H$, $f_B$ and $P^{\mathcal{M}}$ as defined above.

**Construction with arbitrary $|\mathcal{H}| = k$ and $|\mathcal{B}| = m$, $m > k \geq 2$.**

In this second part of the proof, we construct an AI-assisted decision making processes $\mathcal{M}$, with $|\mathcal{H}| = k$ and $|\mathcal{B}| = m$ such that $m > k \geq 2$, such that the decision maker's confidence $f_H$ is monotone, the classifier's confidence $f_B$ is perfectly calibrated and the conditions of Lemma 3 are satisfied.

First, let the space of confidence values be $\mathcal{H} = \{h_i\}_{i \in [k]}$ and $\mathcal{B} = \{b_j\}_{j \in [m]}$, with order $h_i < h_{i+1}$ and $b_i < b_{i+1}$, respectively, and $f_H$, $f_B$ and $P^{\mathcal{M}}$ be such that $P(f_B(Z) = b_j) = 1/m$ and

$$P(H = h_i \mid B = b_j) := P_{X,V}(H = h_i \mid f_B(Z) = b_j) = \begin{cases} \frac{m-j+1}{m} & \text{if } j = i \\ \frac{m-j+1}{m} & \text{if } i = 1, j > k \\ \frac{j-1}{m} & \text{if } j = i+1, j \leq k \\ \frac{j-1}{m} & \text{if } i = k, j > k \\ 0 & \text{otherwise.} \end{cases} \quad (15)$$

Moreover, for each pair of confidence values $(h_i, b_j)$ with positive probability $P(H = h_i, B = b_j)$, we set

$$P(Y = 1 \mid H = h_i, B = b_j) = \begin{cases} P^- & \text{if } j = i \\ P^- & \text{if } i = 1, j > k \\ P^+ & \text{if } j = i + 1, j \leq k \\ P^+ & \text{if } i = k, j > k, \end{cases} \tag{16}$$

as shown in Figure 4 (right). Further, we set the classifier's confidence values $b_j$ to

$$b_j := \frac{m - j + 1}{m} \cdot P^- + \frac{j - 1}{m} \cdot P^+ .$$

Then, it holds that $b_j < b_{j+1}$ and $f_B$ is perfectly calibrated as

$$P(Y = 1 \mid B = b_j) = \begin{cases} P(H = h_j \mid B = b_j) \cdot P^- + P(H = h_{j-1} \mid B = b_j) \cdot P^+ & \text{if } j \leq k \\ P(H = h_1 \mid B = b_j) \cdot P^- + P(H = h_k \mid B = b_j) \cdot P^+ & \text{if } j > k \end{cases}$$

and thus, using the definitions of $P(H \mid B)$ and $P(Y \mid H, B)$, we have that $P(Y \mid B = b_j) = b_j$.

To show that $f_H$ is monotone with respect to the probability that $Y = 1$, first note that $P(H = h_i, B = b_i)$ decreases as $i$ increases and $P(H = h_i, B = b_{i+1})$ increases as $i$ increases. Moreover, further note that $P(Y = 1 \mid H = h_i, B = b_i) = P^- < P(Y = 1 \mid H = h_i, B = b_{i+1}) = P^+$. Hence, for any $i \in \{2, \ldots, k - 1\}$, it readily follows that

$$P(Y = 1 \mid H = h_i) = P^+ \cdot P(B = b_{i+1} | H = h_i) + P^- \cdot P(B = b_i | H = h_i)$$
$$\leq P(Y = 1 \mid H = h_{i+1}),$$

and, for $i = 1$, it is evident that $P(Y = 1 \mid H = h_1) < P(Y = 1 \mid H = h_2)$.

Finally, using Lemma 3 with any choice of confidence values $b = b_j$, $h_i = h_{j-1}$ and $h_j = h_j$ with $j \in \{2, \ldots, k\}$, we have that any monotone AI-assisted decision policy $\pi$ is suboptimal for any $\mathcal{M}$ with $|\mathcal{H}| = k$ and $|\mathcal{B}| = m$, $m > k \geq 2$, and $f_H$, $f_B$ and $P^\mathcal{M}$ as defined above. Here, note that, as we do not fix the exact distributions $P^-$ and $P^+$, the above Lemma applies to infinitely many AI-assisted decision making processes $\mathcal{M}$.

**Construction with $\mathcal{B} \subseteq [0, 1]$ and $|\mathcal{H}| = k$.**

In this last part of the proof, we construct an AI-assisted decision making process $\mathcal{M}$, with $|\mathcal{H}| = k \geq 2$ and $\mathcal{B} \subseteq [0, 1]$, such that the decision maker's confidence function $f_H$ is monotone, the classifier's confidence function $f_B$ is perfectly calibrated and the conditions of Lemma 3 are satisfied.

First, let the space of confidence values be $\mathcal{H} = \{h_i\}_{i \in [k]}$, with order $h_i < h_{i+1}$, the feature space[10] $\mathcal{X} = [0, 1]$, and $f^-, f^+$ be two strictly monotone increasing functions with

$$f^- : [0, 1] \to [0, c) \quad \text{and} \quad f^+ : [0, 1] \to (c, 1], \tag{17}$$

where

$$c = \frac{u(0, 0) - u(1, 0)}{u(1, 1) - u(1, 0) + u(0, 0) - u(0, 1)}. \tag{18}$$

Further, let $Q_{k+1} = \{q_0, q_1, \ldots q_k, q_{k+1}\}$ be a set of quantiles such that $P(X \leq q_j) = j/(k+1)$ for all $j \in \{0, 1, \ldots, k + 1\}$ and thus, we have that, for all $j \in [k + 1]$,

$$\text{for} \quad I_j := (q_{j-1}, q_j], \quad \text{it holds that} \quad P(X \in I_j) = \frac{1}{k + 1}.$$

Now, let $f_H$ and $P^\mathcal{M}$ be such that

$$P_V(H = h_i \mid X, X \in I_j) = \begin{cases} 1/2 & \text{if } i \in \{j - 1, j\} \\ 1 & \text{if } i = j = 1 \text{ or } (i = k \text{ and } j = k + 1) \\ 0 & \text{otherwise}, \end{cases} \tag{19}$$

[10] For a more general feature space $\mathcal{X}$, we can use a mapping $\phi$ of $\mathcal{X}$ to $[0, 1]$. The proof works analogously by substituting $X$ with $\phi(X)$.

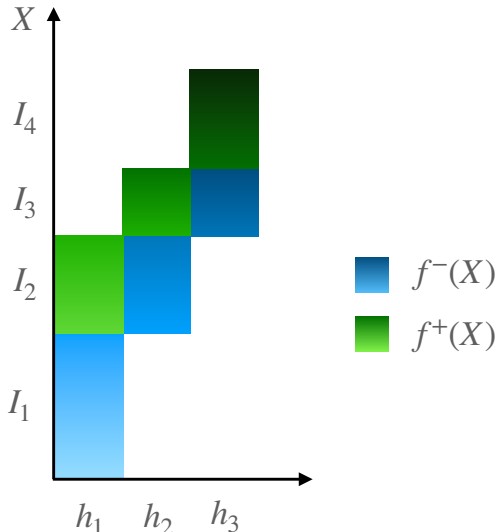

Figure 5: Nonzero values of $P(Y = 1|X, H = h_i, X \in I_j)$ for every $h_i \in \mathcal{H}$, with $|\mathcal{H}| = 3$, and $I_j = (q_{j-1}, q_j]$, with $q_j \in Q_4$ used in the last part of the proof of Theorem 3. Lighter color means lower value of $f^-$ or $f^+$.

and let

$$P(Y = 1 \mid X, H = h_i, X \in I_j) = \begin{cases} f^-(X) & \text{if } j = i \text{ or } (i = j = 1) \\ f^+(X) & \text{if } j = i+1 \text{ or } (i = k \text{ and } j = k+1), \end{cases} \tag{20}$$

as shown in Figure 5. Next, we define

$$f_B(Z) = f_B(X) := P(Y = 1 \mid X) = \begin{cases} f^-(X) & \text{if } X \in I_1 \\ f^+(X) & \text{if } X \in I_{k+1} \\ (f^-(X) + f^+(X))/2 & \text{otherwise,} \end{cases}$$

which, by construction, is perfectly calibrated.

To show that the decision maker's confidence function $f_H$ is monotone with respect to the probability that $Y = 1$, we first note that, using Eq. 19, we have that

$$P(X \in I_j \mid H = h_i) = \begin{cases} 1/2 & \text{if } 1 < i < k \text{ and } j \in \{i, i+1\} \text{ and} \\ 1 & \text{if } i = j = 1 \\ 1 & \text{if } i = k \text{ and } j = k+1 \\ 0 & \text{otherwise.} \end{cases} \tag{21}$$

Hence, using Eq. 21 and the law of total probability, for any $i \in \{2, \ldots, k-2\}$, we have that

$$\begin{aligned} P(Y = 1 \mid H = h_i) &= \frac{1}{2} \left[ P(Y = 1 \mid H = h_i, X \in I_i) + P(Y = 1 \mid H = h_i, X \in I_{i+1}) \right] \\ &\leq \frac{1}{2} \left[ f^-(q_i) + f^+(q_{i+1}) \right] \\ &= \frac{1}{2} \left[ f^- (\inf I_{i+1}) + f^+ (\inf I_{i+2}) \right] \\ &\leq \frac{1}{2} \left[ P(Y = 1 \mid H = h_{i+1}, X \in I_{i+1}) + P(Y = 1 \mid H = h_{i+1}, X \in I_{i+2}) \right] \\ &= P(Y = 1 \mid H = h_{i+1}), \end{aligned}$$

where the inequalities follow from the fact that $f^-$ and $f^+$ are strictly monotone increasing. Corner cases for $i = 1$ and $i = k - 1$ can be shown analogously by further using that $f^-(X) < c < f^+(X)$ for all $X$.

Finally, using Lemma 3 with any choice of confidence values $h_i = h_{j-1}$ $h_j = h_j$, $j \in \{2, \cdots, k-1\}$ and $b = f_B(X)$ with $X \in I_j$, we have that any monotone AI-assisted decision policy $\pi$ is suboptimal for any $\mathcal{M}$ with $|\mathcal{B}| \subseteq [0,1]$ and $|\mathcal{H}| = k$, $k \geq 2$ and $f_H$, $f_B$ and $P^{\mathcal{M}}$ as defined above. $\qquad\square$

## A.3 Proof of Theorem 5

We prove the statement by contraposition. Let $\mathcal{M}$ be an AI-assisted decision making process satisfying Eqs. 2 and 3, with a utility function $u(T, Y)$ satisfying Eq. 1 and let $\mathcal{M}$ be such that $f_B$ satisfies $\alpha$-alignment with respect to $f_H$ and $f_B$ has output space $\mathcal{B} \subseteq [0,1]$. Assume there exists no (near-)optimal monotone AI-assisted decision policy for utility $u$. Thus, there must exist an optimal AI-assisted decision policy $\pi \in \Pi(H, B)$ which is not monotone and has strictly greater expected utility than any monotone policy. However, we show that we can modify $\pi$ to a monotone AI-assisted decision policy $\hat{\pi} \in \Pi(H, B)$ with near-optimal expected utility.

As $\pi$ is not monotone, there must exist confidence values $h_1, h_2 \in \mathcal{H}$, $h_1 \leq h_2$, and $b_1, b_2 \in \mathcal{B}$, $b_1 \leq b_2$, such that
$$\pi(h_1, b_1, w) > \pi(h_2, b_2, w) \quad \text{for some} \quad w \in \mathcal{W}, \tag{22}$$
where $\mathcal{W}$ denotes the space of noise values. In what follows, let $\tilde{\mathcal{W}}_{h_1,b_1}^{(\pi,h_2,b_2)} \subseteq \mathcal{W}$ denote the set containing any such $w$ and let $\tilde{\mathcal{W}}^{(\pi,h_2,b_2)} = \bigcup_{h,b \in \mathcal{H} \times \mathcal{B}} \tilde{\mathcal{W}}_{h,b}^{(\pi,h_2,b_2)}$.

For any confidence value $h', b' \in \mathcal{H} \times [0,1]$, we modify policy $\pi$ to a policy $\hat{\pi}$ as follows. Let $\{\tilde{\mathcal{S}}_h\}_{h \in \mathcal{H}}$ denote the sets satisfying the $\alpha$-alignment condition for $f_B$ with respect to $f_H$ and, given confidence $h'$, let $\hat{b}_{h'}$ denote the smallest confidence value of $f_B$, such that there exist $h \leq h'$ with $P(Y = 1 \mid B = \hat{b}_{h'}, Z \in \tilde{\mathcal{S}}_h) \geq c$, i.e.,
$$\hat{b}_{h'} := \min\{b \in \mathcal{B} \mid P(Y = 1 \mid B = b, Z \in \tilde{\mathcal{S}}_h) \geq c \text{ for } h \leq h'\}. \tag{23}$$

Now, we define a new AI-assisted policy $\hat{\pi}$ from $\pi$ as follows,
$$\hat{\pi}(h', b', w) := \begin{cases} 1 & \text{if } b' \geq \hat{b}_h \text{ and } w \in \bigcup_{h \leq h', b \in [\hat{b}_{h'}, b']} \tilde{\mathcal{W}}^{(\pi,h,b)} \\ 0 & \text{if } b' < \hat{b}_h \text{ and } w \in \bigcup_{h \geq h', b \in [b', \hat{b}_{h'})} \tilde{\mathcal{W}}^{(\pi,h,b)} \\ \pi(h', b', w) & \text{otherwise.} \end{cases} \tag{24}$$

Next, we show that $\hat{\pi}$ is monotone and $\mathbb{E}_{\hat{\pi}}[u(T, Y)] \geq \mathbb{E}_{\pi}[u(T, Y)] + \alpha \cdot a$ for some constant $a$.

**Proof $\hat{\pi}$ is a monotone assisted policy.**

To prove that $\hat{\pi} \in \Pi(H, B)$ is a monotone AI-assisted decision policy, we show that, for all $h', h'' \in \mathcal{H}, b', b'' \in \mathcal{B}$, with $h' \leq h'', b' \leq b''$, it holds that $\tilde{\mathcal{W}}_{h',b'}^{(\hat{\pi},h'',b'')} = \emptyset$. We distinguish between three cases.

— **Case 1**: $b' \geq \hat{b}_{h'}$ and $b'' \geq \hat{b}_{h''}$.

Since $h' \leq h'', b' \leq b''$ and, by definition, $\hat{b}_{h''} \leq \hat{b}_{h'}$ since $h' \leq h''$, we have that
$$\bigcup_{h \leq h', b \in [\hat{b}_{h'}, b']} \tilde{\mathcal{W}}^{(\pi,h,b)} \subseteq \bigcup_{h \leq h'', b \in [\hat{b}_{h''}, b'']} \tilde{\mathcal{W}}^{(\pi,h,b)}.$$

Hence, we can conclude that
$$\hat{\pi}(h', b', w) \leq 1 = \hat{\pi}(h'', b'', w) \text{ for all } w \in \bigcup_{h \leq h'', b \in [\hat{b}_{h''}, b'']} \tilde{\mathcal{W}}^{(\pi,h,b)}. \tag{25}$$

Further, for any other $w \in \mathcal{W} - \bigcup_{h \leq h'', b \in [\hat{b}_{h''}, b'']} \tilde{\mathcal{W}}^{(\pi,h,b)} \subseteq \mathcal{W} - \tilde{\mathcal{W}}_{h',b'}^{(\pi,h'',b'')}$, we have that $\hat{\pi}(h', b', w) = \pi(h', b', w)$ and $\hat{\pi}(h'', b'', w) = \pi(h'', b'', w)$ and, by definition of $\tilde{\mathcal{W}}_{h',b'}^{(\pi,h'',b'')}$, it follows that
$$\hat{\pi}(h', b', w) \leq \hat{\pi}(h'', b'', w) \text{ for all } w \in \mathcal{W} - \bigcup_{h \leq h'', b \in [\hat{b}_{h''}, b'']} \tilde{\mathcal{W}}^{(\pi,h,b)}. \tag{26}$$

From Eqs. 25 and 26, it follows that $\tilde{\mathcal{W}}_{h',b'}^{(\hat{\pi},h'',b'')} = \emptyset$.

— **Case 2**: $b' < \hat{b}_{h'}$ and $b'' \geq \hat{b}_{h''}$.

By definition of $\hat{\pi}$, we have that

$$\hat{\pi}(h',b',w) \leq 1 = \hat{\pi}(h'',b'',w) \text{ for all } w \in \bigcup_{h \leq h'', b \in \left[\hat{b}_{h''},b''\right]} \tilde{\mathcal{W}}^{(\pi,h,b)} \qquad (27)$$

and

$$\hat{\pi}(h',b',w) = 0 \leq \hat{\pi}(h'',b'',w) \text{ for all } w \in \bigcup_{h \geq h', b \in \left[b',\hat{b}_{h'}\right)} \tilde{\mathcal{W}}^{(\pi,h,b)} \qquad (28)$$

Analogously to case 1, since the values of $w$ below are also in $\mathcal{W} - \tilde{\mathcal{W}}_{h',b'}^{(\pi,h'',b'')}$ and $\hat{\pi}$ is equivalent to $\pi$ for these values, we have that

$$\hat{\pi}(h',b',w) \leq \hat{\pi}(h'',b'',w) \text{ for all } w \in \mathcal{W} - \bigcup_{h \leq h'', b \in \left[\hat{b}_{h''},b''\right]} \tilde{\mathcal{W}}^{(\pi,h,b)} - \bigcup_{h \geq h', b \in \left[b',\hat{b}_{h'}\right)} \tilde{\mathcal{W}}^{(\pi,h,b)}$$

$$(29)$$

From Eqs. 27 28 and 29, it follows that $\tilde{\mathcal{W}}_{h',b'}^{(\hat{\pi},h'',b'')} = \emptyset$.

— **Case 3**: $b' < \hat{b}_{h'}$ and $b'' < \hat{b}_{h''}$.

Since $h' \leq h''$, $b' \leq b''$ and, by definition, $\hat{b}_{h''} \leq \hat{b}_{h'}$ since $h' \leq h''$, we have that

$$\bigcup_{h \geq h'', b \in \left[b'',\hat{b}_{h''}\right)} \tilde{\mathcal{W}}^{(\pi,h,b)} \subseteq \bigcup_{h \geq h', b \in \left[b',\hat{b}_{h'}\right)} \tilde{\mathcal{W}}^{(\pi,h,b)}.$$

Hence, we can conclude that

$$\hat{\pi}(h',b',w) = 0 \leq \hat{\pi}(h'',b'',w) \text{ for all } w \in \bigcup_{h \geq h', b \in \left[b',\hat{b}_{h'}\right)} \tilde{\mathcal{W}}^{(\pi,h,b)} \qquad (30)$$

Again analogously to case 1, since the values of $w$ below are also in $\mathcal{W} - \tilde{\mathcal{W}}_{h',b'}^{(\pi,h'',b'')}$ and $\hat{\pi}$ is equivalent to $\pi$ for these values, we have that

$$\hat{\pi}(h',b',w) \leq \hat{\pi}(h'',b'',w) \text{ for all } w \in \mathcal{W} - \bigcup_{h \geq h', b \in \left[b',\hat{b}_{h'}\right)} \tilde{\mathcal{W}}^{(\pi,h,b)} \qquad (31)$$

From Eqs. 30 and 31, it follows that $\tilde{\mathcal{W}}_{h',b'}^{(\hat{\pi},h'',b'')} = \emptyset$.

Note that, we cannot have a case where $b' \geq \hat{b}_{h'}$ and $b'' < \hat{b}_{h''}$, as this would imply $b'' < b'$. Since, in all three possible cases, we have shown that $\tilde{\mathcal{W}}_{h',b'}^{(\hat{\pi},h'',b'')} = \emptyset$, we can conclude that $\hat{\pi} \in \Pi(H,B)$ is monotone.

**Proof $\hat{\pi}$ is near optimal.**

First, we rewrite the inner expectation in Eq. 10 as

$$\mathbb{E}_\pi[u(T,Y) \mid H, B] = \mathbb{E}[u(0,Y) \mid H, B] + (\mathbb{E}[u(1,Y) \mid H, B]$$
$$-\mathbb{E}[u(0,Y) \mid H, B]) \cdot P_\pi(T = 1 \mid H, B).$$

Further, recall that $|\tilde{\mathcal{S}}_h| \geq (1 - \alpha/2)|\mathcal{S}_h|$ for all $h \in \mathcal{H}$ and, for all $h', h'' \in \mathcal{H}$, $h' \leq h''$ and all $b', b'' \in [0,1]$, $b' \leq b''$, we have that

$$P(Y = 1 \mid f_B(Z) = b', Z \in \tilde{\mathcal{S}}_{h'}) - P(Y = 1 \mid f_B(Z) = b'', Z \in \tilde{\mathcal{S}}_{h''}) \leq \alpha \qquad (32)$$

Now, for any $h' \in \mathcal{H}, b' \in \mathcal{B}$, we show an upper bound on $\mathbb{E}_\pi[u(T,Y) \mid H = h', B = b'] - \mathbb{E}_{\hat{\pi}}[u(T,Y) \mid H = h', B = b']$. We distinguish between three cases.

— **Case 1**: $b' \geq \hat{b}_{h'}$ and $P(Y = 1 \mid H = h', B = b') \geq c$.

Using Lemma 2, we have that

$$(\mathbb{E}[u(1, Y) \mid H = h', B = b'] - \mathbb{E}[u(0, Y) \mid H = h', B = b']) \geq 0 \tag{33}$$

Moreover, as $b' \geq \hat{b}_{h'}$, the distribution of positive decisions in $\hat{\pi}$ may also increases for $h', b'$ compared to $\pi$ (see Eq. 24), *i.e.*,

$$P_\pi(T = 1 \mid H = h', B = b') - P_{\hat{\pi}}(T = 1 \mid H = h', B = b') \leq 0$$

Hence, it follows that

$$\mathbb{E}_\pi[u(T, Y) \mid H = h', B = b'] - \mathbb{E}_{\hat{\pi}}[u(T, Y) \mid H = h', B = b']$$
$$= (\mathbb{E}[u(1, Y) \mid H = h', B = b'] - \mathbb{E}[u(0, Y) \mid H = h', B = b']) \tag{34}$$
$$\times (P_\pi(T = 1 \mid H = h', B = b') - P_{\hat{\pi}}(T = 1 \mid H = h', B = b')) \leq 0.$$

— **Case 2**: $b' \geq \hat{b}_{h'}$ and $P(Y = 1 \mid H = h', B = b') < c$.

Since $b' \geq \hat{b}_{h'}$, there exists $h, b \in \mathcal{H} \times \mathcal{B}$, with $h \leq h'$, $b \leq b'$, such that $P(Y = 1 \mid B = b, Z \in \tilde{\mathcal{S}}_h) \geq c$. Moreover, using the definition of $\alpha$-alignment, we have that

$$P(Y = 1 \mid B = b, Z \in \tilde{\mathcal{S}}_h) \leq P(Y = 1 \mid B = b', Z \in \tilde{\mathcal{S}}_{h'}) + \alpha \tag{35}$$

Then, we can use this to lower bound the expected utility of $T = 1$ given $B = b'$ and $Z \in \tilde{\mathcal{S}}_{h'}$ as follows:

$$\mathbb{E}[u(1, Y) \mid B = b, Z \in \tilde{\mathcal{S}}_h] - \mathbb{E}[u(1, Y) \mid B = b', Z \in \tilde{\mathcal{S}}_{h'}]$$
$$= u(1, 1) \cdot (P(Y = 1 \mid B = b, Z \in \tilde{\mathcal{S}}_h) - P(Y = 1 \mid B = b', Z \in \tilde{\mathcal{S}}_{h'})$$
$$+ u(1, 0) \cdot (P(Y = 1 \mid B = b', Z \in \tilde{\mathcal{S}}_{h'}) - P(Y = 1 \mid B = b, Z \in \tilde{\mathcal{S}}_h)) \tag{36}$$
$$\leq (u(1, 1) - u(1, 0)) \cdot \alpha,$$

where the last inequality due to Eq. 35 and the assumption that $u(1, 1) - u(1, 0) > 0$. Analogously, we can also upper bound the expected utility of $T = 0$ given $H = h', B = b'$ and $Z \in \tilde{\mathcal{S}}_{h'}$ as follows:

$$\mathbb{E}[u(0, Y) \mid B = b, Z \in \tilde{\mathcal{S}}_h] - \mathbb{E}[u(0, Y) \mid B = b', Z \in \tilde{\mathcal{S}}_{h'}]$$
$$= u(0, 1) \cdot (P(Y = 1 \mid B = b, Z \in \tilde{\mathcal{S}}_h) - P(Y = 1 \mid B = b', Z \in \tilde{\mathcal{S}}_{h'})$$
$$+ u(0, 0) \cdot (P(Y = 1 \mid B = b', Z \in \tilde{\mathcal{S}}_{h'}) - P(Y = 1 \mid B = b, Z \in \tilde{\mathcal{S}}_h)) \tag{37}$$
$$\geq (u(0, 1) - u(0, 0)) \cdot \alpha,$$

where the last inequality holds due to Eq. 35 and the assumption that $u(0, 1) - u(0, 0) < 0$.

Now, as $P(Y = 1 \mid B = b, Z \in \tilde{\mathcal{S}}_h) \geq c$, by Lemma 2, we have that

$$\mathbb{E}[u(1, Y) \mid B = b, Z \in \tilde{\mathcal{S}}_h] \geq \mathbb{E}[u(0, Y) \mid B = b, Z \in \tilde{\mathcal{S}}_h] \tag{38}$$

Combining Eqs. 36, 37 and 38, we obtain

$$\mathbb{E}[u(1, Y) \mid B = b', Z \in \tilde{\mathcal{S}}_{h'}] + \alpha(u(1, 1) - u(1, 0))$$
$$\geq \mathbb{E}[u(0, Y) \mid B = b', Z \in \tilde{\mathcal{S}}_{h'}] + \alpha(u(0, 1) - u(0, 0)) \tag{39}$$

In addition, note that we have following trivial bound for the expectation when $H = h'$ but $Z \notin \tilde{\mathcal{S}}_{h'}$

$$u(1, 0) \leq \mathbb{E}[u(1, Y) \mid H = h', B = b'] \leq u(1, 1), \tag{40}$$
$$u(0, 1) \leq \mathbb{E}[u(0, Y) \mid H = h', B = b'] \leq u(0, 0) \tag{41}$$

Moreover, since $b' \geq \hat{b}_{h'}$, the distribution of positive decisions in $\hat{\pi}$ may also increase for $h', b'$ compared to $\pi$, *i.e.*,

$$P_\pi(T = 1 \mid H = h', B = b') - P_{\hat{\pi}}(T = 1 \mid H = h', B = b') \leq 0$$

Hence, we have that

$$\mathbb{E}_\pi[u(T, Y) \mid H = h', B = b'] - \mathbb{E}_{\hat{\pi}}[u(T, Y) \mid H = h', B = b']$$
$$\leq (-1) \cdot (\mathbb{E}[u(1, Y) \mid H = h', B = b'] - \mathbb{E}[u(0, Y) \mid H = h', B = b']), \tag{42}$$

where the inequality follows since $\mathbb{E}[u(1,Y) \mid H = h', B = b'] - \mathbb{E}[u(0,Y) \mid H = h', B = b'] \leq 0$ by Lemma 2 as $P(Y = 1 \mid H = h', B = b') < c$.

Finally, combining Eqs. 39, 40, 41 and 42 and using the law of total expectation, we obtain

$$\mathbb{E}_\pi[u(T,Y) \mid H = h', B = b'] - \mathbb{E}_{\hat\pi}[u(T,Y) \mid H = h', B = b']$$
$$\leq (1 - \beta_{(h',b')})(\mathbb{E}[u(0,Y) \mid B = b', Z \in \tilde{\mathcal{S}}_{h'}] - \mathbb{E}[u(1,Y) \mid B = b', Z \in \tilde{\mathcal{S}}_{h'}])$$
$$+ \beta_{(h',b')}(\mathbb{E}[u(0,Y) \mid H = h', B = b'] - \mathbb{E}[u(1,Y) \mid H = h', B = b'])$$
$$\leq (1 - \beta_{(h',b')})\alpha(u(1,1) - u(1,0) + u(0,0) - u(0,1)) + \beta_{(h',b')}(u(0,0) - u(1,0)),$$
$$(43)$$

where $\beta_{(h',b')}$ denotes the probability of $Z \notin \tilde{\mathcal{S}}_{h'}$ given $H = h', B = b'$, i.e., $\beta_{(h',b')} = P(Z \notin \tilde{\mathcal{S}}_{h'} \mid H = h', B = b')$.

— **Case 3**: $b' < \hat{b}_{h'}$.

For all $h, b$, with $h \leq h'$, $b \leq b'$, we have that $P(Y = 1 \mid B = b, Z \in \tilde{\mathcal{S}}_h) < c$. In particular, $P(Y = 1 \mid B = b', Z \in \tilde{\mathcal{S}}_{h'}) < c$. Thus, by Lemma 2,

$$\mathbb{E}[u(1,Y) \mid B = b', Z \in \tilde{\mathcal{S}}_{h'}] < \mathbb{E}[u(0,Y) \mid B = b', Z \in \tilde{\mathcal{S}}_{h'}] \qquad (44)$$

In this case, since $b' < \hat{b}_{h'}$, the distribution of positive decisions in $\hat\pi$ may decrease for $h, b$ compared to $\pi$, i.e.,

$$0 \leq P_\pi(T = 1 \mid H = h, B = b) - P_{\hat\pi}(T = 1 \mid H = h, B = b)$$

Combining Eqs. 44, 40 and 41 and using the law of total expectation, we obtain

$$\mathbb{E}_\pi[u(T,Y) \mid H = h', B = b'] - \mathbb{E}_{\hat\pi}[u(T,Y) \mid H = h', B = b']$$
$$\leq (\mathbb{E}[u(1,Y) \mid H = h', B = b'] - \mathbb{E}[u(0,Y) \mid H = h', B = b']) \cdot 1$$
$$= (1 - \beta_{(h',b')})(\mathbb{E}[u(1,Y) \mid B = b', Z \in \tilde{\mathcal{S}}_{h'}] - \mathbb{E}[u(0,Y) \mid B = b', Z \in \tilde{\mathcal{S}}_{h'}])$$
$$+ \beta_{(h',b')}(\mathbb{E}[u(1,Y) \mid H = h', B = b'] - \mathbb{E}_Y[u(0,Y) \mid H = h', B = b'])$$
$$\leq \beta_{(h',b')}(u(1,1) - u(0,1)),$$
$$(45)$$

where again $\beta_{(h',b')} = P(Z \notin \tilde{\mathcal{S}}_{h'} \mid H = h', B = b')$.

Now, for a fixed $h' \in \mathcal{H}$, since $|\tilde{\mathcal{S}}_{h'}| \geq (1 - \alpha/2)|\mathcal{S}_{h'}|$, we know that $0 \leq \sum_{b\in\mathcal{B}} \beta_{(h',b)} \leq \alpha/2$. Hence, combining Eqs. 34, 43 and 45 from the three cases above, we have that

$$\mathbb{E}_B[\mathbb{E}_\pi[u(T,Y) \mid H = h', B = b']] - \mathbb{E}_B[\mathbb{E}_{\hat\pi}[u(T,Y) \mid H = h', B = b']]$$
$$= \mathbb{E}_B[\mathbb{E}_\pi[u(T,Y) \mid H = h', B = b'] - \mathbb{E}_{\hat\pi}[u(T,Y) \mid H = h', B = b']]$$
$$\leq \max\{\alpha(u(1,1) - u(1,0) + u(0,0) - u(0,1)) + \frac{\alpha}{2} \cdot (u(0,0) - u(1,0)), \ \frac{\alpha}{2} \cdot (u(1,1) - u(0,1))\}$$
$$\leq \alpha \cdot (u(1,1) - u(0,1) + \frac{3}{2} \cdot (u(0,0) - u(1,0))).$$

Finally, since by assumption $\pi$ is optimal, i.e., $\mathbb{E}_\pi[u(T,Y)] = \mathbb{E}_{\pi^*}[u(T,Y)] = \max_{\pi' \in \Pi(H,B)} \mathbb{E}_{\pi'}[u(T,Y)]$, we can conclude by the law of total expectation that

$$\mathbb{E}_{\pi^*}[u(T,Y)] = \mathbb{E}_H \mathbb{E}_B[\mathbb{E}_{Y,\,T\,|\,\pi}[u(T,Y) \mid H, B]]$$
$$\leq \mathbb{E}_{\hat\pi}[u(T,Y)] + \alpha \cdot (u(1,1) - u(0,1) + \frac{3}{2} \cdot (u(0,0) - u(1,0))).$$

This concludes the proof.

### A.4 Proof of Theorem 8

If $f_B$ is $\alpha/2$-multicalibrated with respect to $\{\mathcal{S}_h\}_{h\in\mathcal{H}}$, then, by definition, for any $h \in \mathcal{H}$, there exists $\tilde{\mathcal{S}}_h \subset \mathcal{S}_h$ with $|\mathcal{S}| \geq (1 - \alpha/2)|\mathcal{S}_h|$ such that, for any $b \in [0,1]$, it holds that

$$|P(Y = 1 \mid f_B(Z) = b, Z \in \tilde{\mathcal{S}}_h) - b| \leq \alpha/2.$$

This directly implies that, for any $h', h'' \in \mathcal{H}$ and $b', b'' \in [0, 1]$, we have that

$$P(Y = 1 \mid f_B(Z) = b', Z \in \tilde{\mathcal{S}}_{h'}) - b' - P(Y = 1 \mid f_B(Z) = b'', Z \in \tilde{\mathcal{S}}_{h''}) - b'' \le \alpha \quad (46)$$

and, using linearity of expectation, we further have that

$$P(Y = 1 \mid f_B(Z) = b', Z \in \tilde{\mathcal{S}}_{h'}) - P(Y = 1 \mid f_B(Z) = b'', Z \in \tilde{\mathcal{S}}_{h''}) \le \alpha + b' - b'', \quad (47)$$

showing that, whenever $b' \le b''$, the $\alpha$-alignment condition is met. This proves that $f_B$ is $\alpha$-aligned with respect to $f_H$.

Finally, if $f_B$ is $\alpha/2$-multicalibrated with respect to $\{\mathcal{S}_h\}_{h \in \mathcal{H}}$, then, it is $\alpha/2$-calibrated with respect to any of the sets $\mathcal{S}_h$. Since $\mathcal{Z} = \cup_{h \in \mathcal{H}} \mathcal{S}_h$, this implies that $f_B$ is $\alpha/2$-calibrated with respect to $\mathcal{Z}$. This concludes the proof.

## A.5 Proof of Proposition 1

Given a discretization parameter $\lambda$, Algorithm 1 works with a discretized notion of $\alpha$-multicalibration, namely $(\alpha, \lambda)$-multicalibration:

**Definition 10.** *Let $\mathcal{C} \subseteq 2^{\mathcal{Z}}$ be a collection of subsets of $\mathcal{Z}$. For any $\alpha, \lambda > 0$, confidence function $f_B : \mathcal{Z} \to [0, 1]$ is $(\alpha, \lambda)$-multicalibrated with respect to $\mathcal{C}$ if, for all $S \in \mathcal{C}$, $b \in \Lambda[0, 1]$, and all $\mathcal{S}_{h,\lambda(b)}(g)$ such that $|\mathcal{S}_{h,\lambda(b)}| \ge \alpha\lambda|\mathcal{S}_h|$, it holds that*

$$|\mathbb{E}[f_B(X, H) - P(Y = 1 \mid X, H) \mid (X, H) \in \mathcal{S}_{h,\lambda(b)}]| \le \alpha. \quad (48)$$

Here, we can analogously define a discretized notion of $\alpha$-alignment, namely $(\alpha, \lambda)$-alignment.

**Definition 11.** *For $\alpha, \lambda > 0$, a confidence function $f_B : \mathcal{Z} \to [0, 1]$ is $(\alpha, \lambda)$-aligned with respect to $f_H$ if, for all $h', h'' \in \mathcal{H}$, $h' \le h''$, and all $b', b'' \in \Lambda[0, 1]$, $b' \le b''$, with $|\mathcal{S}_{h',\lambda(b')}| > \alpha/2 \cdot \lambda|S_{h'}|$ and $|\mathcal{S}_{h'',\lambda(b'')}| > \alpha/2 \cdot \lambda|S_{h''}|$, we have*

$$P(Y = 1 \mid (X, H) \in \mathcal{S}_{h',\lambda(b')}) - P(Y = 1 \mid (X, H) \in \mathcal{S}_{h'',\lambda(b'')}) \le \alpha. \quad (49)$$

In what follows, we first show that $(\alpha, \lambda)$-multicalibration with respect to $\{\mathcal{S}_h\}_{h \in \mathcal{H}}$ implies $(2\alpha + \lambda, \lambda)$-alignment with respect to $f_H$.

**Theorem 12.** *For $\alpha, \lambda > 0$, if $f_B$ is $(\alpha, \lambda)$-multicalibrated with respect to $\{\mathcal{S}_h\}_{h \in \mathcal{H}}$, then $f_B$ is $(2\alpha + \lambda, \lambda)$-aligned with respect to $f_H$.*

*Proof.* If $f_B$ is $(\alpha, \lambda)$-multicalibrated with respect to $\{\mathcal{S}_h\}_{h \in \mathcal{H}}$, then, by definition, for all $h \in \mathcal{H}$, $b \in \Lambda[0, 1]$, and all $\mathcal{S}_{h,\lambda(b)}$ such that $|\mathcal{S}_{h,\lambda(b)}| \ge \alpha \cdot \lambda|\mathcal{S}_h|$, it holds that

$$|\mathbb{E}[f_B(X, H) - P(Y = 1 \mid X, H) \mid (X, H) \in \mathcal{S}_{h,\lambda(b)}]| \le \alpha. \quad (50)$$

This directly implies that, for all $h', h'' \in \mathcal{H}, b', b'' \in \Lambda[0, 1]$ with $|\mathcal{S}_{h',\lambda(b')}| \ge \alpha \cdot \lambda|S_{h'}|$ and $|\mathcal{S}_{h'',\lambda(b'')}| \ge \alpha \cdot \lambda|S_{h''}|$, it holds that

$$\begin{aligned}
&\mathbb{E}[f_B(X, H) - P(Y = 1 \mid X, H) \mid (X, H) \in \mathcal{S}_{h'',\lambda(b'')}] \\
&\quad - \mathbb{E}[f_B(X, H) - P(Y = 1 \mid X, H) \mid (X, H) \in \mathcal{S}_{h',\lambda(b')}] \le 2\alpha
\end{aligned} \quad (51)$$

and, using the linearity of expectation, we have that

$$\begin{aligned}
&P(Y = 1 \mid (X, H) \in \mathcal{S}_{h',\lambda(b')}) - P(Y = 1 \mid (X, H) \in \mathcal{S}_{h'',\lambda(b'')}) \\
&\le 2\alpha + \mathbb{E}[f_B(X, H) \mid (X, H) \in \mathcal{S}_{h',\lambda(b')}] - \mathbb{E}[f_B(X, H) \mid (X, H) \in \mathcal{S}_{h'',\lambda(b'')}].
\end{aligned} \quad (52)$$

Whenever $b' \le b''$, due to the $\lambda$-discretization, we have that

$$\mathbb{E}[f_B(X, H) \mid (X, H) \in \mathcal{S}_{h',\lambda(b')}] - \mathbb{E}[f_B(X, H) \mid (X, H) \in \mathcal{S}_{h'',\lambda(b'')}] \le \lambda \quad (53)$$

Hence, we have shown that if $f_B$ is $\alpha$-multicalibrated, then for all $h', h'' \in \mathcal{H}, b', b'' \in \Lambda[0, 1]$ with $|\mathcal{S}_{h',\lambda(b')}| \ge \alpha \cdot \lambda|S_{h'}|$ and $|\mathcal{S}_{h'',\lambda(b'')}| \ge \alpha \cdot \lambda|S_{h''}|$, we have

$$P(Y = 1 \mid (X, H) \in \mathcal{S}_{h',\lambda(b')}) - P(Y = 1 \mid (X, H) \in \mathcal{S}_{h'',\lambda(b'')}) \le 2\alpha + \lambda. \quad (54)$$

Further, note that $(2\alpha + \lambda)/2 \cdot \lambda > \alpha \cdot \lambda$ as $\lambda > 0$. This concludes the proof. $\qquad\square$

Next, we show that, if $f_B$ is $(\alpha, \lambda)$-aligned, then $f_{B,\lambda}$ is $\alpha$-aligned with respect to $f_H$.

**Theorem 13.** *For $\alpha, \lambda > 0$, if $f_B$ is $(\alpha, \lambda)$-aligned with respect to $f_H$, then $f_{B,\lambda}$ is $\alpha$-aligned with respect to $f_H$.*

*Proof.* The proof is similar to the proof of Lemma 1 in Hébert-Johnson et al. [11]. Consider all $\mathcal{S}_{h,\lambda(b)}$ such that $|\mathcal{S}_{h,\lambda(b)}| < \alpha\lambda|\mathcal{S}_h|$. By the $\lambda$-discretization, there are at most $1/\lambda$ such sets, thus, the cardinality of their union is at most $1/\lambda\alpha\lambda|\mathcal{S}_h| = \alpha|\mathcal{S}_h|$. Hence, for all $h \in \mathcal{H}$, there exists a subset $\tilde{\mathcal{S}}_h \subset \mathcal{S}_h$ with $|\tilde{\mathcal{S}}_h| \geq (1-\alpha)|\mathcal{S}_h|$ such that, for all $h', h'' \in \mathcal{H}$, with $h' \leq h''$, and all $b', b'' \in \Lambda[0,1]$, with $b' \leq b''$, it holds that

$$P(Y = 1 \mid (X, H) \in \mathcal{S}_{h',\lambda(b')} \cap \tilde{\mathcal{S}}_{h'}) - P(Y = 1 \mid (X, H) \in \mathcal{S}_{h'',\lambda(b'')} \cap \tilde{\mathcal{S}}_{h''}) \leq \alpha. \quad (55)$$

The $\lambda$-discretization sets all values of $(x, h) \in \mathcal{S}_{h',\lambda(b')}$ to $f_{B,\lambda}(x, h) = \mathbb{E}[f_B(X, H) \mid f_B(X, H) \in \lambda(b')]$. Note that, for $(x, h) \in \mathcal{S}_{h',\lambda(b')}$, $f_{B,\lambda}(x, h) \in \lambda(b')$ and for $(x, h) \in \mathcal{S}_{h'',\lambda(b'')}$, $f_{B,\lambda}(x, h) \in \lambda(b'')$, so it still holds that $\mathbb{E}[f_B(X, H) \mid f_B(X, H) \in \lambda(b')] \leq \mathbb{E}[f_B(X, H) \mid f_B(X, H) \in \lambda(b'')]$. Thus, using Eq. 55, we have that

$$\begin{aligned}
&P(Y = 1 \mid f_B(X, H) = \mathbb{E}[f_B(X, H) \mid (X, H) \in \lambda(b')], (X, H) \in \tilde{\mathcal{S}}_{h'}) \\
&- P(Y = 1 \mid f_B(X, H) = \mathbb{E}[f_B(X, H) \mid (X, H) \in \lambda(b'')], (X, H) \in \tilde{\mathcal{S}}_{h''}) \leq \alpha
\end{aligned} \quad (56)$$

This concludes the proof. $\qquad\square$

Finally, using Theorems 12 and 13, it readily follows that, given a parameter $\alpha'$, the discretized confidence function $f_{B,\lambda}$ returned by Algorithm 1 satisfies $(2\alpha' + \lambda)$-aligned calibration with respect to $f_H$.

### A.6 Proof Theorem 9

We structure the proof in three parts. We first explain the calibration guarantee that UMD provides and how it relates to human-aligned calibration. Then, we derive a lower bound on the size of the subsets $\mathcal{D} \cap \mathcal{S}_h$ so that the discretized confidence function $f_{B,\lambda}$ satisfies $\alpha$-aligned calibration with respect to $f_H$ with high probability. Finally, building on this result, we derive an upper bound on $|\mathcal{D}|$ so that $f_{B,\lambda}$ satisfies $\alpha$-aligned calibration with high probability as long as there exists $\gamma > 0$ so that $P((X, H) \in \mathcal{S}_h) \geq \gamma$ for all $h \in \mathcal{H}$.

**Conditional Calibration implies Human-Aligned Calibration.** Running UMD on a dataset $\mathcal{D} \in (\mathcal{Z} \times \mathcal{Y})^n$, where each datapoint is sampled from $P^{\mathcal{M}}$, guarantees $(\alpha, \xi)$-conditional calibration, a PAC-style calibration guarantee [12]. Given a dataset $\mathcal{D}$, a confidence function $f_B$ satisfies $(\alpha, \xi)$-conditional calibration if, with probability at least $1 - \xi$ over the randomness in $\mathcal{D}$,

$$\forall b \in [0, 1], \quad |P(Y = 1|f_B(X, H) = b) - b| \leq \alpha.$$

This stands in contrast to the definition of $\alpha$-calibration, which requires only that the confidence $f_B(X, H)$ is at most $\alpha$ away from the true probability for $1 - \alpha$ fraction of $\mathcal{Z}$.

Similarly, using an union bound over all $h \in \mathcal{H}$, $(\alpha/2, \xi/|\mathcal{H}|)$-conditional calibration of $f_B$ on each $\mathcal{S}_h, h \in \mathcal{H}$, implies that, with probability at least $1 - \xi$ over the randomness in $\mathcal{D}$, $f_B$ satisfies that

$$\forall h \in \mathcal{H}, \quad \forall b \in [0, 1], \quad |P(Y = 1|f_B(X, H) = b, H = h) - b| \leq \alpha/2. \quad (57)$$

Hence, analogously to the proof of Theorem 8, this implies that, with probability at least $1 - \xi$ over the randomness in $\mathcal{D}$, $f_B$ also satisfies that

$$\begin{aligned}
&\forall h, h' \in \mathcal{H}, h \leq h', \quad \forall b, b' \in \mathcal{G}, b \leq b', \\
&\quad P(Y = 1|f_B(X, H) = b, H = h) - P(Y = 1|f_B(X, H) = b', H = h') \leq \alpha.
\end{aligned} \quad (58)$$

In summary, from Eqs. 57 and 58, we can conclude that $(\alpha/2, \xi/|\mathcal{H}|)$-conditional calibration of $f_B$ on each $\mathcal{S}_h, h \in \mathcal{H}$, implies that, with probability at least $1 - \xi$, $f_B$ satisfies $\alpha$-aligned calibration, where, for all $h \in \mathcal{H}$, we have that $\tilde{\mathcal{S}}_h = \mathcal{S}_h$.

**Lower bound on $|\mathcal{D} \cap \mathcal{S}_h|$ to achieve conditional calibration with UMD.** Running UMD on each partition $\mathcal{D} \cap \mathcal{S}_h$ of $\mathcal{D}$ induced by $h \in \mathcal{H}$ achieves $(\alpha/2, \xi/|\mathcal{H}|)$-conditional calibration as long as each subset $\mathcal{D} \cap \mathcal{S}_h$ of the data is large enough. More specifically, the following lower bound on the size of the subsets $\mathcal{D} \cap \mathcal{S}_h$ readily follows from Theorem 3 in Gupta et al. [12].

**Lemma 4.** *The discretized confidence function $f_{B,\lambda}$ returned by $|\mathcal{H}|$ instances of UMD, one per $\mathcal{S}_h$, is $(\alpha/2, \xi/|\mathcal{H}|)$-conditional calibrated on $\mathcal{S}_h$ for any $\xi \in (0,1)$ if*

$$|\mathcal{D} \cap \mathcal{S}_h| \geq n_{min} := \left( \frac{2 \log \left( \frac{2|\mathcal{H}|}{\xi} \cdot \left\lceil \frac{1}{\lambda} \right\rceil \right)}{\alpha^2} + 2 \right) \cdot \left\lceil \frac{1}{\lambda} \right\rceil \tag{59}$$

*Proof.* Let $B$ denote the number of bins in UMD. Theorem 3 in Gupta et al. [12] states that, if $f_B(X, H)$ is absolutely continuous with respect to the Lebesgue measure[11] and $|\mathcal{D} \cap \mathcal{S}_h| \geq 2B$, then the discretized confidence function output by UMD is $(\epsilon, \xi')$-conditionally calibrated for any $\xi' \in (0,1)$ and

$$\epsilon = \sqrt{\frac{\log(2B/\xi')}{2(\lfloor |\mathcal{D} \cap \mathcal{S}_h|/B \rfloor - 1)}} . \tag{60}$$

Then, for a given $\alpha$, setting $\epsilon = \alpha/2$, $B = \lceil 1/\lambda \rceil$ and $\xi' = \xi/|\mathcal{H}|$, we can solve Eq. 60 for the lower bound on $|\mathcal{D} \cap \mathcal{S}_h| \geq n_{\min}$ with $n_{\min}$ as defined in Eq. 59. $\square$

**Upper bound on $|\mathcal{D}|$ to achieve conditional calibration with UMD.** Suppose $P((X, H) \in \mathcal{S}_h) \geq \gamma$ for all $h \in \mathcal{H}$. When $|\mathcal{H}| \geq 2$, we give an upper bound on $|\mathcal{D}|$ so that with high probability $|\mathcal{D} \cap \mathcal{S}_h| \geq n_{\min}$ for all $h \in \mathcal{H}$.

In the process of sampling $\mathcal{D} \in (\mathcal{Z} \times \mathcal{Y})^n$ from $P^{\mathcal{M}}$, let $R_i^{(h)} = 1$ denote the event that the $i$-th datapoint $(x_i, h_i, y_i)$ has confidence value $h$, *i.e.*, $h_i = h$. Then, we can express $|\mathcal{D} \cap \mathcal{S}_h|$ in terms of random variable $R^{(h)}$, defined as

$$R^{(h)} = \sum_{i=1}^{|\mathcal{D}|} R_i^{(h)} . \tag{61}$$

Since $R_i^{(h)}$ is a Bernoulli-distributed variable with $P(R_i^{(h)}) = P((X, H) \in \mathcal{S}_h)$, the expected value of $R^{(h)}$ is $\mu(h) := \mathbb{E}[R^{(h)}] = P((X, H) \in \mathcal{S}_h) \cdot |\mathcal{D}| \geq \gamma \cdot |\mathcal{D}|$.

Let $|\mathcal{D}| = 2 \cdot |\mathcal{H}| \cdot \log(2/\xi) \cdot 1/\gamma \cdot n_{\min}$, observe that in this case

$$P(R^{(h)} \leq n_{\min}) = P \left( R^{(h)} \leq \frac{\gamma}{2|\mathcal{H}| \cdot \log(2/\xi)} \cdot |\mathcal{D}| \right) .$$

For $|\mathcal{H}| \geq 2$ and $\xi \in (0,1)$, we have $1/(2|\mathcal{H}| \cdot \log(2/\xi)) \in (0,1)$ and we can use a variation of the Chernoff bound to show

$$P(R^{(h)} \leq n_{\min}) \leq P \left( R^{(h)} \leq \frac{1}{2|\mathcal{H}| \cdot \log(2/\xi)} \cdot \mu(h) \right)$$

$$\leq e^{-\mu(h) \left( \frac{2|\mathcal{H}| \cdot \log(2/\xi) - 1}{2|\mathcal{H}| \cdot \log(2/\xi)} \right)^2 \cdot \frac{1}{2}}$$

$$= e^{-\mu(h) \cdot \frac{1}{2} \cdot \left( 1 - \frac{1}{|\mathcal{H}| \cdot \log(2/\xi)} + \frac{1}{(2|\mathcal{H}| \cdot \log(2/\xi))^2} \right)}$$

$$\leq \frac{\xi}{2} \cdot e^{-|\mathcal{H}| \cdot n_{\min} \cdot \left( \frac{1}{2} - \frac{1}{2|\mathcal{H}| \cdot \log(2/\xi)} + \frac{1}{2(2|\mathcal{H}| \cdot \log(2/\xi))^2} \right)} ,$$

where the first and last inequality results from using $\mu(h) > \gamma \cdot |\mathcal{D}|$. We can now use a union bound to obtain a lower bound on the probability that for any $h \in \mathcal{H}$, $|\mathcal{D} \cap \mathcal{S}_h| \leq n_{\min}$, *i.e.*,

$$P(\exists h \in \mathcal{H} : |\mathcal{D} \cap \mathcal{S}_h| \leq n_{\min}) \leq \frac{\xi}{2} \cdot |\mathcal{H}| \cdot e^{-|\mathcal{H}| \cdot n_{\min} \cdot \left( \frac{1}{2} - \frac{1}{2|\mathcal{H}| \cdot \log(2/\xi)} + \frac{1}{2(2|\mathcal{H}| \cdot \log(2/\xi))^2} \right)} \tag{62}$$

One can verify that for $|\mathcal{H}| \geq 2$ and $n_{\min} \geq 1$, we have $P(\exists h \in \mathcal{H} : |\mathcal{D} \cap \mathcal{S}_h| \leq n_{\min}) \leq \frac{\xi}{2}$. Hence, if $|\mathcal{D}| = 2 \cdot |\mathcal{H}| \cdot \log(2/\xi) \cdot 1/\gamma \cdot n_{\min}$, then, for all $h \in \mathcal{H}$, $|\mathcal{D} \cap \mathcal{S}_h| \leq n_{\min}$ with probability $1 - \xi/2$.

---

[11]If $f_B$ is not continuous with respect to the Lebesgue measure (or equivalently put, $f_B$ does not have a probability density function), a randomization trick can be used to ensure that the results of the theorem hold.

Combining this result and Lemma 4, we have that the discretized confidence function $f_{B,\lambda}$ returned by $|\mathcal{H}|$ instances of UMD, one per $\mathcal{S}_h$, is $(\alpha/2, \xi/(2|\mathcal{H}|))$-conditional calibrated on each $\mathcal{S}_h$ with probability at least $1 - \xi/2$ for any $\xi \in (0,1)$ if

$$|\mathcal{D}| = 2 \cdot |\mathcal{H}| \cdot \frac{\log(2/\xi)}{\gamma} \cdot \left( \frac{2 \log\left( \frac{4|\mathcal{H}|}{\xi} \cdot \lceil \frac{1}{\lambda} \rceil \right)}{\alpha^2} + 2 \right) \cdot \left\lceil \frac{1}{\lambda} \right\rceil \tag{63}$$

Finally, using a union bound, we can conclude that $f_{B,\lambda}$ achieves $\alpha$-aligned calibration with respect to $f_H$ with probability at least $1 - \xi$ from

$$|\mathcal{D}| = O\left( |\mathcal{H}| \cdot \frac{\log(|\mathcal{H}|/\xi\lambda)}{\alpha^2 \cdot \lambda \cdot \gamma} \right)$$

samples. This concludes the proof.

# B  Multicalibration Algorithm

In this section, we give a high-level description of the post-processing algorithm for multicalibration introduced by Hébert-Johnson et al. [11]. The algorithm works with a discretization of $[0, 1]$ into uniform sized bins of size $\lambda$, for a $\lambda > 0$. Formally the $\lambda$-discretization of $[0, 1]$, is defined as

**Definition 14** ($\lambda$-discretization [11]). *Let $\lambda > 0$. The $\lambda$-discretization of $[0, 1]$, denoted by $\Lambda[0, 1] = \{\frac{\lambda}{2}, \frac{3\lambda}{2}, \ldots, 1 - \frac{\lambda}{2}\}$, is the set of $1/\lambda$ evenly spaced real values over $[0, 1]$. For $b \in \Lambda[0, 1]$, let*

$$\lambda(b) = [b - \lambda/2, v + \lambda/2) \tag{64}$$

*be the $\lambda$-interval centered around $b$ (except for the final interval, which will be $[1 - \lambda, 1]$).*

It starts by partitioning each subspace $\mathcal{S}_h$ into $1/\lambda$ groups $\mathcal{S}_{h,\lambda(b)} = \{(x, h) \in \mathcal{S}_h \mid f_B(x, h) \in \lambda(b)\}$, with $b \in \Lambda[0, 1]$. Then, it repeatedly looks for a large enough group $\mathcal{S}_{h,\lambda(b)}$ such that the absolute difference between the average confidence value $\mathbb{E}[f_B(X, H) \mid (X, H) \in \mathcal{S}_{h,\lambda(b)}]$ and the probability $P(Y = 1 \mid (X, H) \in \mathcal{S}_{h,\lambda(b)})$ is larger than $\alpha$ and, if it finds it, it updates the confidence value $f_B(x, h)$ of each $(x, h) \in \mathcal{S}_{h,\lambda(b)}$ by this difference. Once the algorithm cannot find any more such a group, it returns a discretized confidence function $f_{B,\lambda}(x, h) = \mathbb{E}[f_B(X, H) \mid f_B(X, H) \in \lambda(b)]$, with $b \in \Lambda[0, 1]$ such that $f_B(x, h) \in \lambda(b)$, which is guaranteed to satisfy $(\alpha + \lambda)$-multicalibration.

Algorithm 1 provides a pseudocode implementation of the overall algorithm. Within the implementation, it is worth noting that the expectations and probabilities can be estimated with fresh samples from the distribution or from a fixed dataset using tools from differential privacy and adaptive data analysis, as discussed in Hébert-Johnson et al. [11].

---

**Algorithm 1** Post-processing algorithm for $(\alpha + \lambda)$-multicalibration

1: **Input:** confidence function $f_B$, parameters $\alpha, \lambda > 0$
2: **Output:** confidence function $f_{B,\lambda}$

3: **repeat**
4:     updated $\leftarrow$ `false`
5:     **for** $\mathcal{S}_h \in \mathcal{C}$ & $b \in \Lambda[0, 1]$ **do**
6:         $\mathcal{S}_{h,\lambda(b)} \leftarrow \mathcal{S}_h \cap \{(x, h) \in \mathcal{Z} \mid f_B(x, h) \in \lambda(b)\}$
7:         **if** $P((X, H) \in \mathcal{S}_{h,\lambda(b)}) < \alpha\lambda \cdot P((X, H) \in \mathcal{S}_h)$ **then**
8:             **continue**
9:         $\bar{b}_{h,\lambda(b)} \leftarrow \mathbb{E}[f_B(X, H) \mid (X, H) \in \mathcal{S}_{h,\lambda(b)}]$
10:        $r_{h,\lambda(b)} \leftarrow P(Y = 1 \mid (X, H) \in \mathcal{S}_{h,\lambda(b)})$
11:        **if** $|r_{h,\lambda(b)} - \bar{b}_{h,\lambda(b)}| > \alpha$ **then**
12:            updated $\leftarrow$ `true`
13:            **for** $(x, h) \in \mathcal{S}_{h,\lambda(b)}$ **do**
14:               $f_B(x, h) \leftarrow f_B(x, h) + (r_{h,\lambda(b)} - \bar{b}_{h,\lambda(b)})$ {project into $[0, 1]$ if necessary}
15: **until** updated $=$ `false`
16: **for** $b \in \Lambda[0, 1]$ **do**
17:     $\bar{b}_{\lambda(b)} \leftarrow \mathbb{E}[f_B(X, H)|f_B(X, H) \in \lambda(b)]$
18:     **for** $(x, h) \in \mathcal{Z} : f_B(x, h) \in \lambda(b)$ **do**
19:         $f_{B,\lambda}(x, h) \leftarrow \bar{b}_{\lambda(b)}$
20: **return** $f_{B,\lambda}$

---

# C   Additional Details about the Experiments

**Transformation of confidence values.** In the Human-AI Interactions dataset, the AI model is a simple statistical model where $b$ is just a noisy average confidence $h$ of an independent set of ca. 50 human labelers on each task instance. Moreover, the confidence values were originally recorded on a scale of $[-1, 1]$, where 1 means complete certainty on the correct true label and $-1$ means complete certainty on the incorrect label. To better match our theoretical framework, we transform all confidence values to a scale of $[0, 1]$, where 1 means complete certainty that the true label $y = 1$ and 0 means complete certainty that the true label is $y \neq 1$. More formally, let $\hat{b}, \hat{h}, \hat{h}_{+\mathrm{AI}} \in [-1, 1]$ be the original confidence values in the dataset, then we obtain $b \in [0, 1]$ via the following transformation:

$$b = \begin{cases} (\hat{b} + 1)/2 & \text{if } y = 1 \\ 1 - (\hat{b} + 1)/2 & \text{if } y = 0, \end{cases}$$

and analogously for $h$ and $h_{+\mathrm{AI}}$.

**Comparing decision policies $\pi_B$, $\pi_H$ and $\pi_{H_{+\mathrm{AI}}}$.** Figure 6 shows the ROC curves for the decision policies $\pi_B$, $\pi_H$ and $\pi_{H_{+\mathrm{AI}}}$ in each of the four tasks in the Human-AI Interactions dataset.

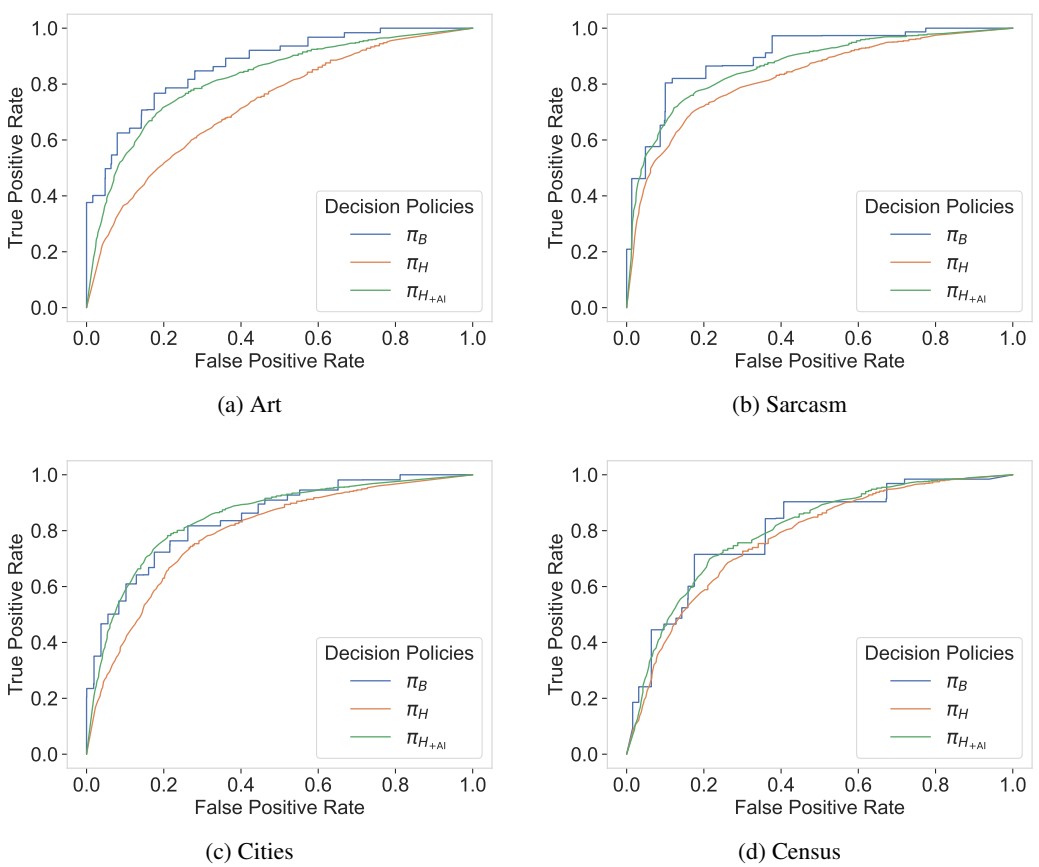

Figure 6: ROC curves for the decision policies $\pi_B$, $\pi_H$ and $\pi_{H_{+\mathrm{AI}}}$.

