# OpenReview forum: "Human-Aligned Calibration for AI-Assisted Decision Making"
_NeurIPS.cc/2023/Conference — NeurIPS 2023 poster_

### Official Review · Reviewer_Q29R · 2023-06-30

**Soundness:** 3 good
**Presentation:** 3 good
**Contribution:** 3 good
**Rating:** 6
**Confidence:** 3

**Summary:**

Communicating model uncertainty to human decision makers is critical for trustworthy deployment of assistive AI systems. Prior work, e.g., Vodrahalli et al 2022, have found that people may respond better to miscalibrated AI confidence scores. What underlies this odd behavior? In this submission, the authors explore this phenomenon, specifically developing a new theoretical framework built on structured causal models. In this framework, the authors find that there indeed are some data distributions wherein calibrated AI confidence scores prohibit humans from uncovering optimal policies by which to make their final decision. The authors then propose to resolve this discrepancy through “human-calibration;” multicalibration is demonstrated as one such method. The authors aim to validate their theoretical insights through empirical studies on the same human data from Vodrahalli et al 2022.

**Strengths:**

A substantive assessment of the strengths of the paper, touching on each of the following dimensions: originality, quality, clarity, and significance. We encourage reviewers to be broad in their definitions of originality and significance. For example, originality may arise from a new definition or problem formulation, creative combinations of existing ideas, application to a new domain, or removing limitations from prior results. You can incorporate Markdown and Latex into your review. See /faq.

This paper has many strengths and was an enjoyable read. The paper was very well-written and incredibly well-motivated. As AI systems increasingly move to real-world applications, it is paramount that they are designed with the human user in mind. The theoretical framework from this paper offers a nice contribution to advance our understanding of the nuances of the way that AI output is communicated to humans to best support decision making.

Some further strengths:
- The specific problem of looking at monotonicity vs. discrepancy from monotonicty in confidence values is interesting and nicely expands on prior work.
- The authors make an excellent point on lines 110-114 that human may take diff decisions with same level of confidence; it is good to see that their model incorporates sources of noise ontop of confidence.
- Nice that their method is claimed to be general to other calibration methods.
- The mathematics seems sound and steps naturally follow from previous; though while I’ve attempted to check through it, there is a possibility I have missed something (hence my own confidence score of 3).


**Weaknesses:**

My biggest misgivings about the paper fall into two categories: 1) the discretization of human confidence, and 2) the empirical validation in Section 6. I will discuss each in turn. I am also concerned that the authors do not adequately discuss limitations. But note this further in the Limitations section of the review.

(1) Discretization of human confidence:
- The authors cite two papers from the neuroscience literature suggesting that humans have “discrete confidence”; however, the studies in these papers looks at quite different decision making tasks than are considered here. The tasks are much lower level and rote, e.g., simple perception and motor tasks, at least from my reading. These are quite different from the kinds of rich, high-level cognitive tasks that involve say, coming up with a treatment plan for a patient. As such, I feel the methods here ought to be able to handle finer-grained representations of human uncertainty/confidence rather than too coarse binning.
- I am very open to discussing this — can the authors shed light on how applicable their theoretical results are to different choices of discretization?
- Specifically on Section 6, I have concerns on the choice of 3 bins. 3 bins are incredibly coarse. Why this number?  For instance, the “mid” confidence level spans a confidence range of 0.5 (out of the normalized 0-1 scale). I recognize that the binning was done here to ensure approximately equal counts across all bins. However, my sense is that this could yield obfuscate some of the nuance in the results? Have the authors considered a finer-grained decomposition, say to 5 or 7 bins? If nothing else, I think this should be discussed in a limitations sections as I imagine this could be an impactful design choice wrt sensitivity in the empirical results.

(2) Empirical validation
- I am struggling to understand the design of the validation in Section 6. What are the baselines? What are the authors trying to show here to validate their theoretical insights (and which aspects of the theoretical insights?) It is not clear to me what is being shown here. When the authors say “participants benefit from AI advice” in Table 1 [line 333], how is this different from the original paper from which the data is sourced?
- The authors note that the results (which per the above, I’d be keen to hear explicated more – what is the expected behavior of these graphs, relative to what baselines?) are weaker for the Census task. Do the authors have a sense for why this may be?
- The authors grouped the results of cases where participants were told that an AI system had provided the confidence with those told a human had (footnote 9 on pg 8). Is this valid? Could have influenced participants’ uptake of the knowledge? Have the authors explored how results stand up if they decompose / keep those groups separate? Do trends still hold?

**Questions:**

- Many of my questions are listed in the Weaknesses section. In particular, it would be great if the authors could clearly spell out which parts of their experiments (Section 6) validate which parts of their theory. I am open to reconsidering some of my critiques from the Weakness section if this is clarified a bit better.
- Why 8 or 10 bins? [line 298]
- Could you expand please on the confounding factors in the data from other counties? I am not entirely sure why you threw away that data? [lines 287-288] My understanding is this was because those participants were told the AI had a different accuracy? This may be worth exploring (perhaps in future work) – how do your theoretical results hold up across wider swaths of data?)
- How does this relate to RLHF? Please spell out the connection a bit more, or o.w. I actually think it’s unnecessary to mention here. [lines 73-74]


**Limitations:**

The authors do not adequately discuss limitations, nor any potential negative societal impact. I would be keen for the authors to provide their thoughts on these two points during the rebuttal/discussion period.

---

> ### Author Rebuttal · Authors · 2023-08-06
>
> We thank the reviewer for their careful and insightful comments, which will help improve our paper. Please, find a point-by-point response below.
>
> **[Discretization of human confidence]** As long as the human confidence scores are represented using a discrete ordered set, our theoretical results are applicable. For example, one may decide to use a finer-grained representation of human uncertainty/confidence using a **finite** set of real values ${0, 0.05, 0.1, …, 0.95, 1.0}$. However, the finer-grained the representation of human confidence, the greater the amount of data that is needed for (i) reliably estimating the probabilities depicted in Figure 2 and the level of misalignment, as measured by EAE and MAE, and (ii) running the multicalibration algorithms described in Section 5 with theoretical guarantees, as discussed in lines 247-250 and 266-268. In our experiments, we used $3$ bins to have sufficient data for (i).
>
> **[Why 8 or 10 bins? [line 298]]** For each task, we chose the number of bins based on the amount of available data. To avoid any misunderstanding, we have re-generated all figures and tables using 8 bins for all tasks and did not find any major difference (please, refer to the companion pdf submitted with the rebuttal for the new table and figures of the “Art” and “Sarcasm” task using 8 bins). We will update the figures and table and include error bars, in the revised version of the paper.
>
> **[Section 6: what are the baselines?]** The baselines are (i) decision policies $\pi_B$ constructed by thresholding the classifier's confidence values and (ii) decision policies $\pi_H$ constructed by thresholding the humans' confidence values _before_ observing the classifier's confidence values. In our experiments, we compare these baselines with (iii) decision policies $\pi_{H_{+AI}}$ constructed by thresholding the humans' confidence values _after_ observing the classifier's confidence values.
>
> **[Section 6: what do we show to validate the theoretical insights?]** In the context of the specific dataset we used in our experiments, our results support our theory as follows:
> (i) Figure 2 and Table 1 (left and middle columns) demonstrate that AI models often violate the necessary condition for human-alignment, as defined in Section 4.
> (ii) Figure 3 suggests that, whenever they receive AI advice, humans make decisions following a monotone decision policy. This supports our hypothesis in Section 2.
> (iii) Table 1 (right column) shows that, **across tasks**, the level of misalignment of the AI model is inversely correlated with the performance of AI-assisted human decision makers. This supports Theorems 3 and 5.
>
> The original paper from which the dataset we used was sourced did not demonstrate any of the above (i-iii). In this context, we would also like to acknowledge that our experiments do not demonstrate that, **for a fixed task**, the degree of misalignment of the AI model is inversely correlated with the performance of AI-assisted human decision makers. The reason being that this pre-existing dataset does not contain human predictions before and after multicalibrating an AI model. The only way we see to address this limitation is to run a human subject study, however, this would require significant funding and time and, given the on-going discussions about the potential of human+AI collaboration, we think the NeurIPS community may benefit more from the theoretical framework we introduce in our paper if published early. In future work, we are planning to run such a study, which we think would be very worthy on its own.
>
> **[Section 6: expected behavior, performance in the Census task weaker?]** It is often assumed that AI-assisted human decision makers ($\pi_{H_{+AI}}$) should outperform both human decision makers ($\pi_{H}$) and AI decision makers ($\pi_{B}$). Our experiments support the former—AI-assisted human decision makers outperform human decision makers consistently across all tasks—but not the latter—AI-assisted human decision makers only outperforms AI decision makers in a single task ("Cities") out of four. We cannot really tell why the level of human misalignment of the AI model is so high in the “Census” task.
>
> **[Section 6: AI advice vs human advice]** In the dataset we used in our experiments, a participant was told that the advice came from AI or from humans at random in all tasks, i.e., there was randomization with respect to this factor of variation. Therefore, whenever we compare the level of misalignment, the level of miscalibration and AUC across tasks, we are essentially controlling for this factor of variation. Unfortunately, if we keep the above groups separately, we do not have sufficient data to estimate reliably the level of misalignment and miscalibration. In this context, we would also like to clarify that, while we focus on AI-assisted human decision making, our theoretical results do not depend on who (AI model or another human) gives the _advice_. As long as the advice comes in the form of confidence values, our results are valid. We will include a discussion about this in the revision.
>
> **[Data from other countries]** We did not include data from other countries because participants were told that the model had different accuracies and they only performed two of the tasks ("Art" and "Sarcasm"). However, we agree with the reviewer that it would be important to replicate our experimental findings using other datasets as well as carry out human subject studies in future work.
>
> **[RLHF]** Since the term human alignment is used widely in RLHF, typically in the context of human preferences, we think it is necessary to clarify that our notion of human-alignment is very different.
>
> **[Limitations & potential negative societal impact]** In the revised version of the paper, we will include an explicit section to discuss limitations, including the experiments' limited scope, the setup's simplicity, and the limitations discussed in lines 347-353.

---

> > ### Comment · Reviewer_Q29R · 2023-08-11
> > **Response to Authors - Adjusted Score**
> >
> > Dear Authors,
> >
> > Thank you for your thorough rebuttal. I appreciate the effort you went through to produce clarifying results around discretization, and the care in which you responded to my (many) points.
> >
> > Your clarification on what aspects of the empirical results validated theoretical results, as well as clarifying the baselines and set-up, lead me to adjust me score. I have edited accordingly.
> >
> > I would encourage the authors to update the next iteration of the manuscript to more clearly highly the correspondence of empirical to theoretical results as they have here.
> >
> > I appreciate that the authors will update their Limitations section as well. I think the authors did an excellent job delineating the distinction between generalizing results across vs. for a fixed task. The latter of which would be great to add to the Limitations (or generally the Discussion); the authors' notes on why a human study is warranted, but til sufficient for future work, is well justified here. Having this in the updated manusript text would be useful.
> >
> > Thank you as well for clarifying the choice of 3 bins. The lines referenced in the text do provide further detail on this point; however, I think the text (and reader) would benefit from including a higher-level note about the relationship between the amount of discretization and data needed for the estimation.
> >
> > "In this context, we would also like to clarify that, while we focus on AI-assisted human decision making, our theoretical results do not depend on who (AI model or another human) gives the advice. As long as the advice comes in the form of confidence values, our results are valid." <-- This is a powerful point as well. I look forward to the authors' promised expanded discussion in their revision.

---

> > > ### Author Response · Authors · 2023-08-15
> > >
> > > Thank you for your response and your effort to review our work. We are glad that we could clarify your questions and appreciate the adjustment of the score.

---

### Official Review · Reviewer_TBZV · 2023-07-06

**Soundness:** 3 good
**Presentation:** 3 good
**Contribution:** 2 fair
**Rating:** 6
**Confidence:** 3

**Summary:**

This paper proposes a causal model of AI-assisted decision-making that considers both the classifier’s and decision maker’s confidence scores. The authors demonstrate theoretically why it may be challenging for the human to identify the optimal decision-making policy even if the classifier is perfectly calibrated. The authors propose a remedy by providing theoretical results using multicalibration. The authors then provide experimental results to measure misalignment and miscalibration in real-world datasets.

**Strengths:**

- The paper was generally well written and lays out the proposed causal model of AI-assisted decision-making in a step-by-step manner, making it easy to follow
- The paper covers an important, growing topic on AI-assisted decision-making and highlights the need to consider notions of both human and AI calibration


**Weaknesses:**

My main concern with this paper is the lack of empirical results that demonstrate an *improvement* in $\mathbb{E}_\pi [u(T,Y)]$ (i.e., show results that the AUC would improve in the H+AI condition as compared to $\pi_B$ or $\pi_H$, which is a commonly used definition of complementarity) as stated as a contribution in L62. While they provide a theoretical result (Thrm 5) to guarantee near-optimality under $\alpha$-alignment, I do not believe the authors implemented multicalibration (the proposed solution in Section 5) in their empirical results. The ideal result would be to show that without multicalibration, human decision makers more often do not achieve complementary performance, as shown in Table 1 last column, and then this could be corrected by introducing some version of multicalibration. Such an implementation would suggest how the theoretical result could be translated to practice and could be very impactful in the human-AI decision-making literature.

**Questions:**

Major question:
Could the authors please address the concerns with the empirical results above?

Minor questions:
- In L121, you state that "allow the classifier’s confidence to depend on the decision maker’s confidence". What is a realistic example of a classifier that takes in decision maker confidence?
- Why is Eq. 4 important to measure?
- Are there any potential issues that can be introduced if you represent human confidence scores in different ways (e.g., binning into low-med-high or using real numbers like 0.7/0.8)?
- Can you provide error bars for Table 1?
- The term "misalignment" is introduced in the empirical results and seemed to be an important quantity of evaluation, could the authors clarify why the term was not more formally incorporated in the earlier sections?

**Limitations:**

Yes

---

> ### Author Rebuttal · Authors · 2023-08-06
>
> We thank the reviewer for their careful and insightful comments, which will help improve our paper. Please, find a point-by-point response below.
>
> **[Empirical results]** We acknowledge that our experiments do not empirically demonstrate that, **for a fixed task**, there is an improvement in $E_{\pi}[u(T, Y)]$ before and after alignment. However, they empirically demonstrate that, **across tasks**, the degree of alignment of the AI model correlates with the performance of $\pi_{H_{+AI}}$ as compared to $\pi_H$ and $\pi_B$. As the reviewer may guess, the reason why we could not show the former is because, in our experiments, we used an observational dataset, gathered in a previous work, and this dataset does not contain human predictions before and after multicalibrating $f_{B}$ (i.e., the AI). The only way we see to address this limitation is to run a human subject study, however, this would require significant funding and time and, given the on-going discussions about the potential of human+AI collaboration, we think the NeurIPS community may benefit more from the theoretical framework we introduce in our paper if published early. That being said, we are planning to run such a study, which we think would be very worthy on its own, in future work. To avoid any misunderstanding, we will rewrite L62 and explicitly discuss this limitation in the revised version of the paper.
>
> **[Realistic example]** In our SCM, we allowed the classifier's confidence to depend on the decision maker's confidence because, to achieve human-alignment via multicalibration as described in Section 5, the classifier needs to have access to the decision maker's confidence. However, Theorems 3 and 5 still hold if the classifier does not depend on the classifier's confidence. Further, we would like to clarify that, as of today, existing classifiers do not (typically) depend on the decision maker's confidence. However, looking into the future, our work questions this status quo by showing that, by allowing the model’s confidence to depend on the decision maker's confidence, a decision maker may end up taking decisions with higher utility. We will add a discussion about this in the revised version of the paper.
>
> **[Eq. 4]**  Eq. 4 is important because it indicates whether the decision maker is actually taking decisions using a monotonic policy. More specifically, if Eq. 4 does not hold, then the policy is not monotonic.
>
> **[Human confidence scores]** As long as the human confidence scores are represented using a discrete ordered set, we cannot think of any algorithmic nor theoretical issues. For example, one may decide to use the set ${0, 0.1, 0.2, …, 0.9, 1.0}$ or any other finite set of real values. That being said, the larger the set of human confidence scores used in the representation, the greater the data and computational requirements to achieve human-alignment via multicalibration, as discussed in lines 247-250 and 266-268.
>
> **[Error bars]** We will provide error bars for Table 1 in the revised version of the paper.
>
> **[Term misalignment]** The misalignment measures EAE and MAE introduced in Section 6 quantify the average and maximum value of the left hand side of Eq. 6 in Section 4 given $\tilde{S}_h = S_h$, whenever those values are positive.  We will make this connection more explicit in the revised version of the paper.

---

> > ### Comment · Reviewer_TBZV · 2023-08-16
> >
> > Thanks to the authors for carefully responding to my concerns. I have read the author responses to all of the reviews and feel that if the paper was sufficiently revised to more clearly state the intended scope of the work and the set of future work that are implied by the theoretical contributions, the paper would be beneficial to the NeurIPS community. I will update my score accordingly.

---

> > > ### Author Response · Authors · 2023-08-21
> > >
> > > Thank you for your response and your effort to review our work. We appreciate the increase of the score.

---

### Official Review · Reviewer_oDhU · 2023-07-07

**Soundness:** 3 good
**Presentation:** 3 good
**Contribution:** 4 excellent
**Rating:** 7
**Confidence:** 3

**Summary:**

This paper theoretically analyzes human decision making in the presence of AI-provided predictions and confidence estimates. They do this by defining a causal model to characterize AI-assisted decision making along with a sensible assumptions about a rational decision-maker's policy: namely that their probability of making a positive classification is monotone in the confidence values. Under this assumption, the paper shows that the traditionally accepted idea of "calibrated" confidence is not enough for a rational decision maker to make optimal decisions. However, a new notion of "human-alignment" does provide this guarantee, and in fact is achievable via the well-known idea of multicalibration. The paper also presents some experiments to validate their assumptions and to argue that  human-aligned confidences might lead to better decision making.

**Strengths:**

This paper approaches the question of AI-assisted decision making from first principles, allowing them to ground their assumptions explicitly and provide a compelling theoretical analysis under their model. The flavour of results in this paper is quite interesting and their techniques build upon and connect a some rich areas of theoretical research. Lastly, the exposition of the paper is quite easy to read, with a good breakdown of high-level ideas and detailed maths in the appendix.

I believe this work would be of notable interest to several communities at NeurIPS.

**Weaknesses:**

To me, the weakest part of this paper is the experimental section, which in my opinion doesn't add a huge amount of insight into the results. I like the idea of empirically demonstrating the monotonicity assumption, however the interplay between misalignment and the human decision-makers' ability to perform the task well is mentioned like an afterthought. It would be amazing to see a scientifically valid empirical claim that human alignment leads to better performance with AI-assisted decision making. This is likely a significant task to carry out, and it is fine to claim as beyond the scope of a theoretical paper. However, it would make this an extremely strong submission.

I would also like to see a discussion of the limitations of this work.

**Questions:**

In several of the experiments, the human performance is worse than that of the model itself. This could be due to several factors, but one reason could be that these humans were not experts at the specified task. In terms of your SCM, they aren't using a lot of latent information V that is hidden to the model. Is there a way to include a "competence/expertise" parameter in this SCM to model, and even possibly correct, a non-expert decision maker so that their performance exceeds that of the AI system?

**Limitations:**

Limitations of the work not discussed. What are the limits of this papers' ideas in terms of both theoretical follow-up and real-world usage?

---

> ### Author Rebuttal · Authors · 2023-08-06
>
> We thank the reviewer for their careful and insightful comments, which will help improve our paper. Please, find a point-by-point response below.
>
> **[Empirical validation that human-alignment leads to higher human performance]** We acknowledge that our experiments do not empirically demonstrate that, **for a fixed task**, there is an improvement in $E_{\pi}[u(T, Y)]$ before and after alignment. However, they empirically demonstrate that, **across tasks**, the degree of alignment of the AI model correlates with the performance of $\pi_{H_{+AI}}$ as compared to $\pi_H$ and $\pi_B$. As the reviewer may guess, the reason why we could not show the former is because, in our experiments, we used an observational dataset, gathered in a previous work, and this dataset does not contain human predictions before and after multicalibrating $f_{B}$ (i.e., the AI). The only way we see to address this limitation is to run a human subject study, however, this would require significant funding and time and, given the on-going discussions about the potential of human+AI collaboration, we think the NeurIPS community may benefit more from the theoretical framework we introduce in our paper if published early. That being said, we are planning to run such a study, which we think would be very worthy on its own, in future work. To avoid any misunderstanding, we will explicitly discuss this limitation in the revised version of the paper.
>
> **[Limitations]** In addition to the limitation just discussed in the previous answer and the limitations/future work discussed in lines 347-353, we acknowledge that our theoretical results apply only to a simple setting where both the outcome variable and the decision are binary. It would be very interesting to analyze settings with categorical, or even real valued, outcome variables and decisions as well as sequential settings. One of the main challenges would be to identify which natural conditions utility functions may satisfy in such settings. In our revised version of the paper, we will include an explicit section describing the above limitations.
>
> **[Competence/expertise in the SCM]** We agree that, in our experiments, humans were unlikely to be _experts_ at the specified tasks and were not using additional information. However, note that our results do not depend on the human's expertise and, given enough historical data about a human decision maker with any level of expertise, one can align any black box model to that decision maker via multicalibration. In this context, we would also like to clarify that our theoretical and empirical results suggest that, if the classifier is sufficiently human-aligned and the decision task reduces to predicting the value of the outcome, a non-expert decision maker can achieve higher performance after observing the classifier's human-aligned confidence.

---

> > ### Comment · Reviewer_oDhU · 2023-08-16
> >
> > Thank you for your helpful clarifications!

---

### Official Review · Reviewer_iDbU · 2023-07-08

**Soundness:** 3 good
**Presentation:** 2 fair
**Contribution:** 3 good
**Rating:** 5
**Confidence:** 3

**Summary:**

The paper studies a binary decision-making process augmented by AI assistance. The decision maker optimizes a utility function satisfying certain natural conditions. The decision rule depends on the decision maker's (DM) confidence and a classifier's confidence. It is shown that there is a lack of alignment between DM's and classifier's confidences; the optimal rule might not have a natural property of placing more trust in predictions with higher confidence values. The paper then proves that a (near-)optimal decision rule with such a property exists if both confidences do not appear contradictory, with high probability, to the rational decision maker, a property dubbed "human-alignment". The paper connects this property with multicalibration algorithms for the classifier's confidence function. The paper illustrates the approach on real human data.

**Strengths:**

The paper introduces a theoretical framework to study optimal decision-making in binary problems, where the decision rule depends on the DM's and classifier's confidence.  The results like this are needed to understand the decision-making process when interacting with AI, particularly now, when strong AI systems are making their way into everyday lives.
The theory was illustrated with a real-life data experiment, where the paper asks questions about misalignment, the influence of AI confidence on human confidence values, and a comparison between decision policies.

**Weaknesses:**

The setup considered in the paper is very simple. It deals with binary decision problems and the experiment provided in the paper actually does not use any modern ML classifier (see footnote 8).

Section 6 would significantly improved in terms of clarity. In particular,
* In the data description part, it is not clear how the labels are defined. For instance, for the "Cities," does y is random (line 283), or does it encode the correct answer (in which case, e.g., h='low' would correspond to a human having low confidence in the correct answer). In a similar vein, it would help to explicitly define how are empirical values of $\mathbb P(Y=1|(X,Y)\in S_{h,\lambda(b)})$ estimated (line 319; what exact data are used).
* Figures 2 and 3 use an "alignment violations" term, which is not defined.
* Figures 2 and 3 should be better explained: each figure contains a set of four plots, each with from 8 to 10 sets of triplets of bars. In particular, the monotonicity in lines 318, 327 should be discussed in a more reader-friendly way.
* The data used in the experiment were aggregated in various ways. The claims like "support for our hypothesis that (rational) decision-makers implement monotone AI-assisted decisions policies" should probably be treated with caution.

**Questions:**

It would be interesting to see how the Authors envision future work in the context of LLM-based systems, multi-categorical or sequential
decision-making.

**Limitations:**

The paper does not have a limitations section. Natural limitations include the setup's simplicity or the experiments' limited scope.

---

> ### Author Rebuttal · Authors · 2023-08-06
>
> We thank the reviewer for their careful and insightful comments, which will help improve our paper. Please, find a point-by-point response below.
>
> **[Very simple setup]** Our theoretical results in Sections 3-4 and the multicalibration algorithms in Section 5 do not make any assumption about the classifier. Therefore, as long as the classifier provides a confidence value, we think our results are applicable to any modern ML classifier (e.g.,  deep neural networks). The reason why we did not use a modern ML classifier in our experiments is because we used an observational dataset gathered in a previous work and this work used the ML classifier described in footnote 8. The only way we see to address this limitation is to run a human subject study, however, this would require significant funding and time and, given the on-going discussions about the potential of human+AI collaboration, we think the NeurIPS community may benefit more from the theoretical framework we introduce in our paper if published early.
>
> **[Label definition]**  For the task “Cities” and “Art”, there are four label values overall but, for each instance, there are only two choices of label values. Therefore, in our experiments, we map each instance’s label values to label values {0, 1} at random. For example, for the “Cities” task, for an instance with label choice “Los Angeles” vs. “San Francisco”, we map “Los Angeles” to label 0 and “San Francisco” to label 1. For this instance, if the true label is “Los Angeles”, we have Y = 0. If h = ’low’, the human decision maker is confident that the true label is not “San Francisco”. Here, note that we did not have control on the way label values were initially assigned since we used observational data gathered in a previous work.
>
> **[Estimation of $P(Y=1 | (X,Y) \in S_{h, \lambda(b)})$]** This probability is computed as the empirical average of the outcome variable $Y$ across instances where the human confidence $H = h$ and the discretized classifier's confidence $B = b$,  where we used $8$ or $10$ bins.
>
> **[Definition of alignment violations]** There is an alignment violation between confidence values $h, b$ and $h’, b’$, with $h\leq h’$ and $b\leq b’$, whenever $P(Y =1 | (X,Y) \in S_{h,\lambda(b)}) - P(Y =1 | (X,Y) \in S_{h’,\lambda(b')}) > 0$. We will clarify this in the revised version of the paper.
>
> **[Explanations of Figures 2 and 3]** In the revised version of the paper, we will improve the explanations of Figures 2-3 particularly regarding the monotonicity in lines 318 and 327.
>
> **[Empirical support for monotone AI-assisted decision policies]**  Following the reviewer's advice, in the revised version of the paper, we will emphasize that, since the data we used in the experiment were aggregated in various ways, our conclusions should be treated with caution.
>
> **[Other settings]** We think our theoretical results may be a good starting point to study settings with multi-categorical outcome variables and decisions. However, one of the main challenges would be to identify which natural conditions utility functions may satisfy in such settings. A theoretical analysis of AI-assisted sequential decision making seems significantly more challenging—multicalibration in sequential settings is an open area of research—but our theoretical results may still be a useful starting point. A theoretical analysis of LLM-based assisted decision making is rather open and it would be certainly a landmark result. We will include a discussion about this in the revised version of the paper.
>
> **[Limitations]** In the revised version of the paper, we will include an explicit section to discuss limitations, including the setup's simplicity, the experiments' limited scope and the limitations discussed in lines 347-353.

---

> > ### Comment · Reviewer_iDbU · 2023-08-18
> >
> > Thank you for the clarification and overall discussion. I am inclined to keep the current rating.

---

### Author Rebuttal · Authors · 2023-08-06

We thank the reviewers for their careful and insightful comments, which will help improve our paper.

In addition to the point-by-point responses to each individual reviewer below, please, find attached a pdf containing updated figures for the “Art” and “Sarcasm” tasks as well as an updated Table 1 where we have used 8 bins for the discretization of the AI model's confidence in all tasks (in the submitted version, we have used 10 bins for the tasks "Art" and "Sarcasm"). We did not find any major difference in the results based on this change.

We will provide error bars for Table 1 in the revised version of the paper.

---

### Decision · Program_Chairs · 2023-09-21

**Decision:**

Accept (poster)

**Comment:**

The paper provides a new theoretical framework for calibrating predictions of AI models for optimal decision making by downstream agents. There was broad consensus amongst the reviewers on the strengths and weaknesses of the proposed work. On the positive side, all the reviewers appreciated the importance of the problem and the elegance of the proposed approach in addressing it for arbitrary binary classifiers. On the negative side, the reviewers had 2 major remarks: one on the limitation of the proposed framework to settings with binary outcomes/targets, and the second on the limitations of the experiments to simple experimental setups which comes across a missed opportunity for demonstrating the broad importance and utility of the proposed framework. These limitations are understandable for a first work, as also reflected in the fact that the overall recommendation of all reviewers was towards acceptance.